# Trust but Verify: Adaptive Conditioning for Reference-Based Diffusion Super-Resolution via Implicit Reference Correlation Modeling

**Yuan Wang**[1,2,*]  **Yuhao Wan**[1]  **Siming Zheng**[2]  **Bo Li**[2]  **Qibin Hou**[1]  **Peng-Tao Jiang**[2,†]

[1]Nankai University      [2]vivo BlueImage Lab, vivo Mobile Communication Co., Ltd.

[†]Corresponding author: `pt.jiang@vivo.com`

## Abstract

Recent works have explored *reference-based super-resolution* (RefSR) to mitigate hallucinations in diffusion-based image restoration. A key challenge is that real-world degradations make correspondences between low-quality (LQ) inputs and reference (Ref) images unreliable, requiring adaptive control of reference usage. Existing methods either ignore LQ–Ref correlations or rely on brittle explicit matching, leading to over-reliance on misleading references or under-utilization of valuable cues. To address this, we propose **Ada-RefSR**, a single-step diffusion framework guided by a *"Trust but Verify"* principle: reference information is leveraged when reliable and suppressed otherwise. Its core component, **Adaptive Implicit Correlation Gating (AICG)**, employs learnable summary tokens to distill dominant reference patterns and capture implicit correlations with LQ features. Integrated into the attention backbone, AICG provides **lightweight, adaptive** regulation of reference guidance, serving as a built-in safeguard against erroneous fusion. Experiments on multiple datasets demonstrate that Ada-RefSR achieves a strong balance of fidelity, naturalness, and efficiency, while remaining robust under varying reference alignment. Code and models are available at this url.

## 1 Introduction

Diffusion-based single-image super-resolution (SISR) methods (Wang et al., 2024b; Wu et al., 2024b) have recently attracted significant attention, owing to the strong generative priors they inherit from large-scale text-to-image models. While capable of producing visually pleasing results, these methods often suffer from hallucinations and out-of-distribution artifacts (Aithal et al., 2024), fabricating or omitting details due to the inherently ill-posed nature of SISR.

To address these challenges, researchers have explored *reference-based super-resolution* (RefSR) (Jiang et al., 2021; Zhang et al., 2023a), which incorporates an auxiliary reference image (Ref) alongside the LQ input to provide complementary high-frequency details for guided reconstruction. The effectiveness of RefSR depends on establishing meaningful cross-modal correspondences between LQ and Ref features so that useful information can be transferred. Yet, when LQ images are heavily degraded, matching becomes unreliable, rendering it critical to regulate how much the model relies on references versus its intrinsic SR capability.

To balance this reliance, some strategies have been proposed, most of which can be interpreted as weighting mechanisms, differing mainly in how the weights are determined. PFStorer (Varanka et al., 2024), designed for face SR, employs a global learnable vector that uniformly controls the reference branch across inputs, but fails to adapt to different alignment qualities (e.g., well-aligned vs. poorly matched pairs). ReFIR (Guo et al., 2024), on the other hand, generates spatial gates from token-wise similarity maps, but these explicit correlation metrics are easily disrupted by noise and long-tail distributions (where majority identical tokens dominate the computation over minority critical ones). In practice, such weighting-based approaches often suffer from two issues (cf. Fig. 1): (1) The weighting mechanism may result in excessive or insufficient reliance on references—over-injecting reference cues (Fig. 1 top row, green-boxed column) or under-utilizing them (Fig. 1 top

---

[*]Work done when interning at vivo.

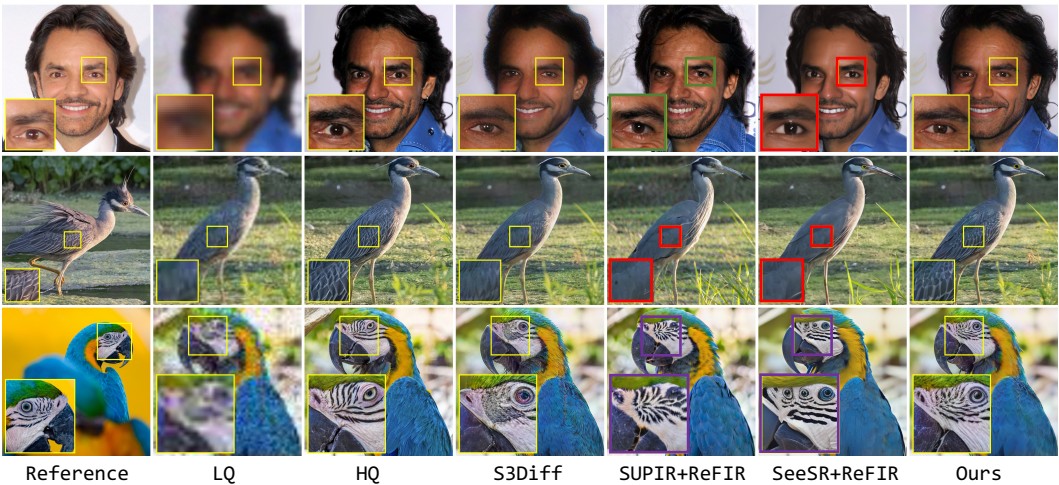

Reference    LQ    HQ    S3Diff    SUPIR+ReFIR    SeeSR+ReFIR    Ours

Figure 1: Visual comparisons: S3Diff is a single-image generation network, and ReFIR is the current SOTA for reference based restoration. Our method not only outperforms ReFIR in leveraging reference details but also shows stronger robustness against degradations than S3Diff.

row, red-boxed column; middle row, red-boxed regions); (2) Semantic mismatches caused by noisy explicit correspondences (e.g., duplicated bird eyes in Fig. 1 bottom row, purple-boxed region, due to ReFIR's erroneous matching of regions to bird eyes), which introduce unnatural regions. Fig. 2(a) and Fig. 2(b) illustrate these behaviors: PFStorer's global gating fails to adapt to alignment quality, while ReFIR's explicit correlation maps are disrupted by noise, leading to unreliable gating.

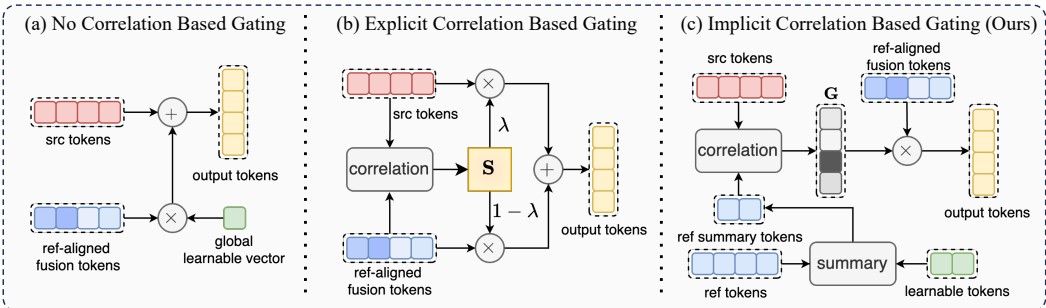

Figure 2: Comparison of gating strategies. Prior methods either use non-interactive global gating (PFStorer) or depend on explicit token correlations that are easily disturbed by noise (ReFIR). In contrast, we introduce implicit, correlation-driven gating that selectively activates useful reference cues. "src tokens" denote main-branch features, "ref tokens" are raw reference features, and "ref-aligned fusion tokens" are reference features aligned to the source space through feature fusion. Here, **G** denotes the per-token gating values generated by our AICG, while **S** represents the full token-to-token similarity map used in explicit correlation-based gating methods like ReFIR.

Our key insight addresses the challenges outlined above. Reference fusion should maximize the use of reliable regions while suppressing misleading ones to prevent unnatural artifacts. Following a *"Trust but Verify"* paradigm, the model first **trusts** reference regions likely aligned with the LQ input to enhance detail recovery, and then **verifies** and attenuates unreliable regions to reduce distortions. This approach naturally addresses both the over/under-reliance on references and semantic mismatches in weighting-based methods.

Motivated by this principle, we propose **Ada-RefSR**, a single-step reference-based diffusion framework. Its core component, **Adaptive Implicit Correlation Gating (AICG)** (Fig. 2(c)), employs learnable summary tokens to implicitly model latent correspondences between LQ and reference features. These tokens function as a compact intermediary, extracting salient reference cues while filtering out semantically inconsistent information. When applied after reference injection, AICG

adaptively modulates reference contributions, providing on-the-fly correction of erroneous feature integration. This mechanism preserves the fidelity and naturalness of the generated content, while enhancing robustness in retrieval-augmented scenarios characterized by variable reference quality. Our contributions can be summarized as follows:

- **AICG: Adaptive Implicit Correlation Gating.** We propose **AICG**, a lightweight implicit correlation gating module that directly addresses a key challenge in RefSR: how to reliably use reference information to restore LQ inputs degraded by real-world artifacts. By reusing existing projections in the attention module and introducing only a few learnable summary tokens, AICG implicitly models LQ–Ref correlations while adding negligible computational overhead.

- **Ada-RefSR: Strong generalization, robustness, and speed.** Built upon AICG, **Ada-RefSR** achieves stable reference-based enhancement across diverse tasks and degradation scenarios. Its single-step diffusion design provides over $30\times$ speedup compared to multi-step RefSR baselines, enabling fast and robust SR in both aligned and mismatched reference conditions.

## 2 RELATED WORK

### 2.1 DIFFUSION MODEL FOR IMAGE RESTORATION

Diffusion-based super-resolution has rapidly advanced in recent years, with a variety of models exploring different ways to exploit text-to-image generative priors. StableSR (Wang et al., 2024b) introduced a time-aware encoder to balance fidelity and naturalness, while DiffBIR (Lin et al., 2024) first restores base results before refining details via diffusion. PASD (Yang et al., 2024) incorporated a degradation removal module with pixel-aware cross-attention, and SeeSR (Wu et al., 2024b) enhanced semantic understanding through a degradation-aware prompt extractor. Most of these approaches rely on multi-step denoising, which is computationally expensive. To improve efficiency, recent works turn to single-step paradigms: SinSR (Wang et al., 2024c) distills a one-step model from a multi-step teacher, OSEDiff (Wu et al., 2024a) simplifies ControlNet with KL regularization, S3Diff (Zhang et al., 2024) adopts degradation-guided LoRA, and PISA-SR (Sun et al., 2025) decouples pixel and semantic-level objectives. Despite these advances, diffusion-based SR still suffers from intrinsic hallucination, where spurious structures emerge due to insufficient constraints.

### 2.2 REFERENCE BASED IMAGE SUPER RESOLUTION

RefSR enhances SISR by leveraging auxiliary reference images. Early works such as SRNTT (Zhang et al., 2019) treat it as neural texture transfer, while TTSR (Yang et al., 2020) introduces cross-scale attention for texture alignment. To preserve domain-specific features, Hu et al. (2020) utilized 3D facial priors as reference guidance. MASA-SR (Lu et al., 2021) adapts spatial features to handle distribution discrepancies, C2-Matching (Jiang et al., 2021) exploits contrastive correspondence with knowledge distillation, and DATSR (Cao et al., 2022) applies deformable attention for multi-scale transfer. LMR (Zhang et al., 2023a) further extends the paradigm to multi-reference settings, and PFStore (Varanka et al., 2024) employs learnable vectors to balance reference influence. Despite these advances, most methods assume manually specified registration between LQ and reference images, limiting real-world applicability. ReFIR (Guo et al., 2024) moves toward retrieval-augmented RefSR, but its training-free design and fixed reference weighting restrict adaptability in practice.

## 3 METHODOLOGY

In this section, we introduce our reference-based super-resolution framework named **Ada-RefSR**, which aims to recover high-quality images from low-quality (LQ) observations $\mathbf{X}_{lq}$ by adaptively leveraging an external reference image $\mathbf{X}_{ref}$, either retrieved from a large-scale database or directly provided by the user. Our method consists of two key components: (1) a reference attention module for injecting fine-grained details, and (2) a correlation-aware adaptive gating mechanism to balance reference utilization and input consistency. Figure 3 illustrates the overall pipeline.

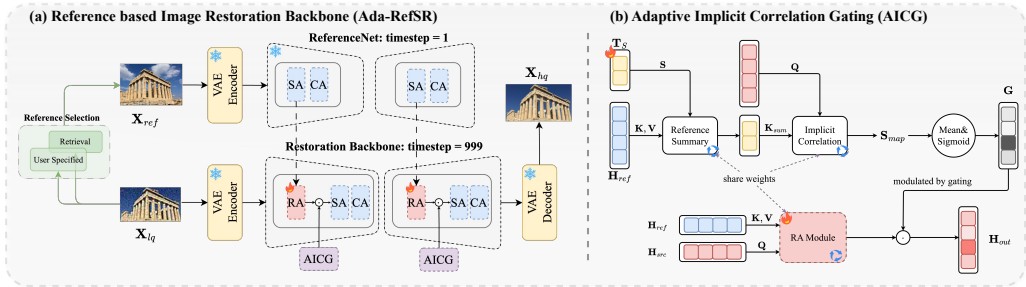

Figure 3: Overview of our framework. It comprises two components: (a) a reference-based restoration backbone, and (b) a correlation-aware adaptive gating mechanism.

## 3.1 TRUST: DIRECT REFERENCE FEATURE INJECTION

**Maximizing Reference Utilization in the Trust Phase.** The "trust" phase aims to maximize the utilization of reference features, ensuring that all potential LQ–reference matches are captured for detail enhancement. The intention is to ensure that no potentially useful correspondence is discarded at this stage, even at the risk of introducing mismatches. We build on a single-step diffusion SR backbone that already provides strong fidelity for $\mathbf{X}_{lq}$, and augment it with reference-guided detail extraction without upfront filtering. Existing methods are insufficient: IP-Adapter-style approaches (Ye et al., 2023) transfer only global identity, missing local matches, while ControlNet-style methods (Zhang et al., 2023b) discard valid features under spatial misalignment (Hu et al., 2023). To overcome these limitations, we employ ReferenceNet (Hu et al., 2023) instantiated with SD-Turbo (Sauer et al., 2023) and fix its timestep to 1, enabling full-stage, fine-grained reference feature extraction.

**Injecting Reference Features via Attention.** To leverage these extracted features, we employ a Reference Attention (RA) module (Fig. 3(a)). RA is initialized by copying the backbone's self-attention weights, with backbone features as queries and ReferenceNet features as key-value pairs—allowing every LQ query to selectively draw from all reference cues and directly realize the "trust" objective. Let the backbone feature at a given layer be $\mathbf{H}_{src} \in \mathbb{R}^{L_{src} \times d}$, and the corresponding ReferenceNet feature be $\mathbf{H}_{ref} \in \mathbb{R}^{L_{ref} \times d}$. We define the attention projections as:

$$\mathbf{Q} = \mathbf{H}_{src}\mathbf{W}_Q, \quad \mathbf{K} = \mathbf{H}_{ref}\mathbf{W}_K, \quad \mathbf{V} = \mathbf{H}_{ref}\mathbf{W}_V, \tag{1}$$

where $\mathbf{W}_Q, \mathbf{W}_K, \mathbf{W}_V \in \mathbb{R}^{d \times d}$ are learnable projection matrices. The vanilla RA is then computed as:

$$\mathbf{H}_{out} = \text{ZeroLinear}\Big(\text{Softmax}\Big(\frac{\mathbf{Q}\mathbf{K}^\top}{\sqrt{d}}\Big)\mathbf{V}\Big) + \mathbf{H}_{src}, \tag{2}$$

where ZeroLinear stabilizes training in early stages and prevents disruption of the original Stable Diffusion features (Rombach et al., 2021). The above method extracts detail-rich, spatially meaningful features via ReferenceNet and fuses them with LQ backbone features through RA module without requiring spatial alignment. Residual connections further preserve the prior knowledge of the original super-resolution network.

While vanilla RA effectively injects reference details, its indiscriminate fusion often causes local semantic inconsistencies and unnatural textures (Fig. 4). Attention map visualizations reveal the root cause: the "trust" phase inevitably allows LQ regions to pull features from semantically mismatched areas in the reference (e.g., non-eye regions borrowing eye features). These issues underscore the need for a complementary verification phase that selectively regulates reference influence.

## 3.2 VERIFY: ADAPTIVE IMPLICIT CORRELATION GATING

An effective way to realize this verification is through a gating mechanism, mediating between *baseline restoration fidelity* and *reference-guided enhancement*. However, existing gating strategies remain limited. PFStore employs a global learnable vector, disregarding LQ–Ref feature correlations and thus failing to adapt to varying alignment quality. ReFIR constructs spatial gates via token-wise similarity but suffers from two inherent drawbacks: (1) explicit correspondences are highly

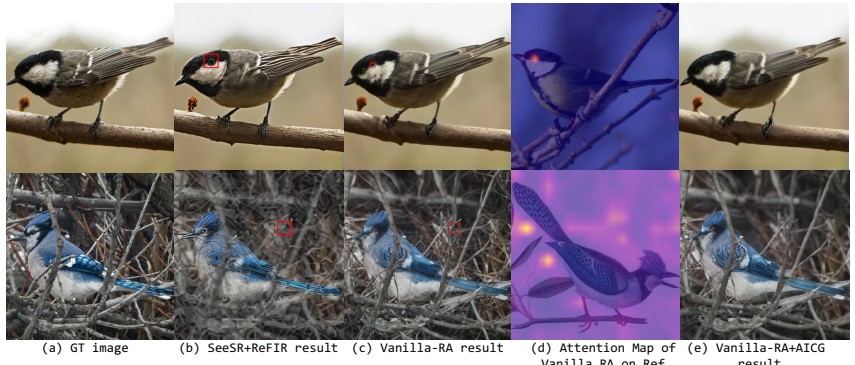

(a) GT image    (b) SeeSR+ReFIR result    (c) Vanilla-RA result    (d) Attention Map of Vanilla RA on Ref    (e) Vanilla-RA+AICG result

Figure 4: Vanilla RA and ReFIR both introduce artifacts by overusing irrelevant reference regions (left: duplicate eye; right: cluttered background). Our gating mechanism suppresses these activations, leading to more natural results. Zoom in for better visualization.

susceptible to noise (Bolya et al., 2023) and long-tail biases; (2) reliance on a fixed user-specified weight (0.5) prevents adaptation to varying alignment quality. These weaknesses manifest clearly in our comparative experiments (Tab. 2).

To address these limitations and fulfill the "Verify" phase of our *"Trust but Verify"* paradigm, we propose an Adaptive Implicit Correlation Gating (**AICG**) mechanism (Fig. 3(b)), which implicitly models LQ–Ref correlation to adaptively regulate reference guidance. While inspired by the high-level idea of learnable tokens in DETR (Carion et al., 2020), AICG differs fundamentally: Instead of serving as decoding queries, our tokens summarize reference features to provide implicit, token-wise reliability estimates for gating. This design suppresses noise and mismatches while retaining essential structures. Importantly, AICG remains lightweight, introducing only a few learnable tokens and reusing existing projections and intermediate variables within the RA module.

**Reference Summarization via Learnable Tokens.** We introduce a set of learnable summary tokens $\mathbf{T}_S \in \mathbb{R}^{M \times d}$ that compactly summarize the reference features. First, we map them into the key space using the reference-attention projection:

$$\mathbf{S} = \mathbf{T}_S \mathbf{W}_K, \tag{3}$$

where $\mathbf{W}_K$ is the key projection matrix in Eq. 1. We then compute the correlation between summary tokens and reference keys:

$$\mathbf{K}_{sum} = \text{Softmax}\left(\frac{\mathbf{S}\mathbf{K}^\top}{\sqrt{d}}\right)\mathbf{K} \in \mathbb{R}^{M \times d}, \tag{4}$$

where $\mathbf{K} \in \mathbb{R}^{L_{ref} \times d}$ is the reference key sequence in Eq. 1. Intuitively, $\mathbf{K}_{sum}$ captures the most informative textures and foreground structures in a compact $M$-token form ($M \ll L_{ref}$), incurring negligible additional cost.

**Implicit Reference Correlation based Gating.** Next, we compute the attention between LQ queries and the summarized reference:

$$\mathbf{S}_{map} = \text{Softmax}\left(\frac{\mathbf{Q}\mathbf{K}_{sum}^\top}{\sqrt{d}}\right) \in \mathbb{R}^{L_q \times M}, \tag{5}$$

where $\mathbf{Q}$ are the LQ query sequence in Eq. 1. We aggregate the logits across $M$ summary tokens and attention heads to form a scalar gating value for each LQ token:

$$\mathbf{G} = \sigma\left(\frac{1}{M}\sum_{j=1}^{M}[\mathbf{S}_{map}]_{:,j}\right) \in \mathbb{R}^{L_q \times 1}, \tag{6}$$

where $\sigma(\cdot)$ denotes the sigmoid function. Finally, the vanilla RA output is adaptively modulated as:

$$\mathbf{H}_{out} = \text{ZeroLinear}\Big(\mathbf{G} \odot \text{RA}(\mathbf{H}_{src}, \mathbf{H}_{ref})\Big) + \mathbf{H}_{src}, \tag{7}$$

where $\odot$ denotes element-wise scaling along the token dimension.

**Discussion.** Our adaptive gating mechanism embodies the **"Trust but Verify"** principle in reference-based diffusion SR. Instead of indiscriminately injecting reference information, it adaptively regulates contributions—trusting semantically aligned regions while suppressing mismatches (see Fig.4, Fig.6)—thereby mitigating hallucinations and artifacts. In addition, learnable summary tokens provide a compact interface between large-scale reference features and LQ queries, reinforcing robustness and making implicit correlation modeling both adaptive and reliable (see Fig. 7).

## 4 EXPERIMENTS

### 4.1 EXPERIMENTAL SETTINGS

**Training Datasets and Implementation Details.** We adopt S3Diff (Zhang et al., 2024) as our baseline, fine-tuning only the new reference attention module while freezing other components to preserve pre-trained super-resolution priors. Training data include: (1) synthetic SR datasets (DIV2K, DIV8K, Flickr2K), with $1024 \times 1024$ regions cropped into two $512 \times 512$ sub-regions (HQ and Ref), Ref images augmented via random rotation (details in appendix A.1); (2) a face reference matching dataset (Li et al., 2022) with predefined HQ-Ref triplets. LQ images for all datasets use the RealSRGAN degradation pipeline (Wang et al., 2021). To improve robustness, 20% of HQ-Ref pairs in synthetic datasets are randomly replaced with irrelevant samples. Training uses two NVIDIA A40 GPUs, Adam optimizer (Kingma & Ba, 2014), initial learning rate $5 \times 10^{-5}$, batch size 16, and 11K iterations for convergence. The training objective follows the original design of S3Diff, combining an $L_2$ reconstruction loss with perceptual and GAN losses, with detailed formulation provided in appendix B.

**Evaluation datasets and Metrics.** To evaluate our method's reference effectiveness in dense matching, we use datasets CUFED5 (Zhang et al., 2019) and WRSR (Jiang et al., 2021), processed following ReFIR's protocol (Guo et al., 2024). Following Varanka et al. (2024), we test 40 human identities: 5 non-overlapping high-quality (HQ) pairs per identity, yielding 162 test pairs (all 512×512). To verify category-based retrieval enhancement, we construct a self-collected bird dataset ( 8,460 HQ images; 66 as HQ images, others as retrieval database, all 512×512; details in appendix A.2). Consistent with prior works, we use four reference-based metrics (PSNR, SSIM (Wang et al., 2004), LPIPS (Zhang et al., 2018), FID (Heusel et al., 2017)) and three no-reference metrics (NIQE (Zhang et al., 2015), CLIPIQA (Wang et al., 2023), MUSIQ (Ke et al., 2021)).

**Compared Methods.** To evaluate our method in mitigating hallucinations in diffusion-based models, we compare against RealSRGAN (Wang et al., 2021), PISA-SR (Sun et al., 2025), S3Diff (Zhang et al., 2024), OSEDiff (Wu et al., 2024a), and large-model reference SR approaches like SeeSR+ReFIR and SUPIR+ReFIR (Guo et al., 2024) on general scene datasets. For dense-scene reference datasets, we add C2Matching (Jiang et al., 2021) and DATSR (Cao et al., 2022) to evaluate reference generation. For face datasets, we further compare with DMDNet (Li et al., 2022), CodeFormer (Zhou et al., 2022), FaceMe (Liu et al., 2025) and InstantRestore (Zhang et al., 2025) to validate face-specific reference SR performance. To ensure fairness, all use a single reference image, with the number of reference images set to 1 for all methods.

### 4.2 COMPARATIVE RESULTS

**Quantitative Comparison.** We present quantitative results on four datasets with varying reference matching degrees, as shown in the table. Several observations can be made:

- **Overall performance.** Our method achieves the best results across most reference-based metrics (PSNR, SSIM, LPIPS, FID), consistently surpassing the baseline S3Diff and other diffusion based model. On no-reference metrics (NIQE, MUSIQ, CLIPQA), it attains comparable scores, showing that reference cues are effectively exploited without compromising perceptual naturalness.

- **Comparison with reference-based methods.** Compared with ReFIR, our approach delivers substantially higher fidelity, confirming its advantage in structural and textural restoration. Notably, SUPIR+ReFIR underperforms our method, suggesting that ReFIR lacks general adaptability across different SR backbones. Moreover, our method outperforms

Table 1: Algorithm performance comparison across multiple datasets. The best performance is highlighted in **red bold**, the second-best in **blue bold**, and rows corresponding to "Ours" are shaded with a light background. Notation: S-x denotes inference with x time steps; * indicates methods that use reference images.

| Dataset | Method | PSNR ↑ | SSIM ↑ | LPIPS ↓ | FID ↓ | NIQE ↓ | MUSIQ ↑ | CLIPQA ↑ |
|---------|--------|--------|--------|---------|-------|--------|---------|----------|
| CUFED5 | C2-Matching* | **21.0199** | 0.5303 | 0.8040 | 292.5921 | 9.5425 | 16.8868 | 0.1227 |
| | DATSR* | **21.0196** | 0.5306 | 0.8071 | 292.7726 | 9.6305 | 16.8736 | 0.1269 |
| | Real-ESRGAN-S1 | 20.3084 | **0.5543** | 0.3697 | 175.9116 | 3.8861 | 64.7123 | 0.4214 |
| | S3Diff-S1 | 20.4594 | 0.5234 | **0.3544** | 160.1337 | **3.7096** | 63.8519 | **0.5238** |
| | OSEDiff-S1 | 18.8748 | 0.4935 | 0.3669 | 150.7504 | 4.0005 | 65.2446 | 0.5077 |
| | PISA-SR-S1 | 20.3371 | 0.5286 | 0.3707 | 164.6927 | 4.1992 | 64.2840 | 0.5043 |
| | SUPIR+ReFIR*-S50 | 18.9994 | 0.4711 | 0.4359 | 151.2700 | 4.2745 | 62.7384 | 0.4235 |
| | SeeSR+ReFIR*-S50 | 20.2169 | 0.5255 | 0.3452 | **137.3693** | **3.7426** | **70.5375** | **0.5538** |
| | **Ours-S1*** | 20.4843 | **0.5461** | **0.2894** | **127.8945** | 3.8701 | **67.9070** | 0.5174 |
| WRSR | C2-Matching* | **22.8766** | 0.5784 | 0.7471 | 153.1606 | 9.0121 | 19.9772 | 0.1527 |
| | DATSR* | **22.9017** | **0.5820** | 0.7583 | 153.5259 | 9.3999 | 20.1124 | 0.1613 |
| | Real-ESRGAN-S1 | 22.1491 | **0.5974** | 0.3631 | 97.9142 | **3.6976** | 64.5176 | 0.6076 |
| | S3Diff-S1 | 21.9122 | 0.5620 | 0.3542 | 63.8237 | 4.4171 | 63.8237 | 0.6149 |
| | OSEDiff-S1 | 21.3263 | 0.5603 | **0.3261** | 71.5148 | 3.7364 | **71.1935** | **0.6899** |
| | PISA-SR-S1 | 21.8813 | 0.5682 | 0.3383 | 62.9958 | 4.0359 | 69.5187 | 0.6407 |
| | SUPIR+ReFIR*-S50 | 21.0134 | 0.5378 | 0.3949 | 76.4993 | **3.6407** | 70.1242 | 0.6219 |
| | SeeSR+ReFIR*-S50 | 21.8334 | 0.5673 | 0.3435 | **61.9597** | 3.8919 | **72.1520** | **0.7120** |
| | **Ours-S1*** | 21.9722 | 0.5777 | **0.3061** | **53.2811** | 3.7429 | 69.8608 | 0.6612 |
| Bird | Real-ESRGAN-S1 | **25.0809** | 0.7044 | 0.5306 | 178.8849 | 8.8496 | 29.3109 | 0.3571 |
| | S3Diff-S1 | 24.8422 | **0.7106** | 0.2903 | 60.9452 | 5.3195 | 73.1125 | 0.8285 |
| | OSEDiff-S1 | 23.7861 | 0.7086 | 0.2949 | **49.1911** | **4.9796** | **74.7567** | **0.8451** |
| | PISA-SR-S1 | 24.4891 | 0.7003 | **0.2816** | 67.9959 | **4.8513** | 74.3633 | 0.8373 |
| | SUPIR+ReFIR*-S50 | 23.8344 | 0.6974 | 0.3811 | 94.9298 | 5.4422 | 63.6740 | 0.6478 |
| | SeeSR+ReFIR*-S50 | 23.7211 | 0.6985 | 0.2945 | 52.4038 | 5.6182 | **75.1056** | **0.8617** |
| | **Ours-S1*** | **25.2998** | **0.7292** | **0.2536** | **36.4249** | 5.1771 | 72.0039 | 0.8303 |
| Face | Code-former | 26.2677 | 0.7082 | 0.3798 | 99.6599 | 5.4105 | 63.9886 | 0.5192 |
| | DMDNet* | **26.8993** | **0.7472** | 0.2316 | 56.6262 | 4.6030 | 72.9495 | 0.6395 |
| | Real-ESRGAN-S1 | 26.1678 | 0.7142 | 0.5113 | 162.4299 | 8.9504 | 26.4788 | 0.3630 |
| | S3Diff-S1 | 26.6881 | 0.7344 | 0.2189 | 51.1219 | 5.0599 | 74.1095 | 0.6963 |
| | OSEDiff-S1 | 25.5302 | 0.7434 | 0.2263 | 55.0428 | 4.9550 | 71.7721 | 0.6459 |
| | PISA-SR-S1 | 26.2433 | 0.7254 | 0.2126 | **48.9154** | 4.5423 | **75.9855** | **0.7191** |
| | SUPIR+ReFIR*-S50 | 25.7231 | 0.7062 | 0.2623 | 57.1432 | **3.9686** | 73.0854 | 0.6260 |
| | SeeSR+ReFIR*-S50 | 26.3191 | 0.7375 | 0.2293 | 57.2910 | 5.2885 | **75.4230** | **0.7236** |
| | FaceMe*-S50 | 26.6591 | 0.7245 | 0.2500 | 54.3278 | 4.7295 | 72.4144 | 0.6483 |
| | InstantRestore*-S1 | 26.2245 | 0.7350 | **0.2070** | 51.2283 | 5.9473 | 69.7209 | 0.5726 |
| | **Ours-S1*** | **27.1271** | **0.7523** | **0.1749** | **42.7042** | **4.4880** | 74.3828 | 0.6453 |

DMDNet and InstantRestore across all metrics, highlighting its effectiveness in face reference super-resolution.

- **Task-specific observations.** In scene-level reference SR, traditional approaches (e.g., C2-Matching, DATSR) yield higher PSNR/SSIM, while diffusion-based methods emphasize perceptual realism. On the WRSR benchmark, our method also demonstrates a strong fidelity–perception balance under real-world degradations. On bird and face benchmarks, our method consistently outperforms existing approaches in reference-based metrics, with only minor variations in no-reference metrics due to imperfect LQ–Ref alignment. Notably, in face reference SR, our method surpasses recent SOTA approaches (Faceme, InstanceRestore) by leveraging the baseline SR prior via reference attention, while AICG implicitly filters irrelevant reference regions to prevent misleading feature injection.

Overall, our method shows clear advantages in reference-based metrics while maintaining strong competitiveness in no-reference evaluations, demonstrating its effectiveness in utilizing reference information without compromising perceptual quality.

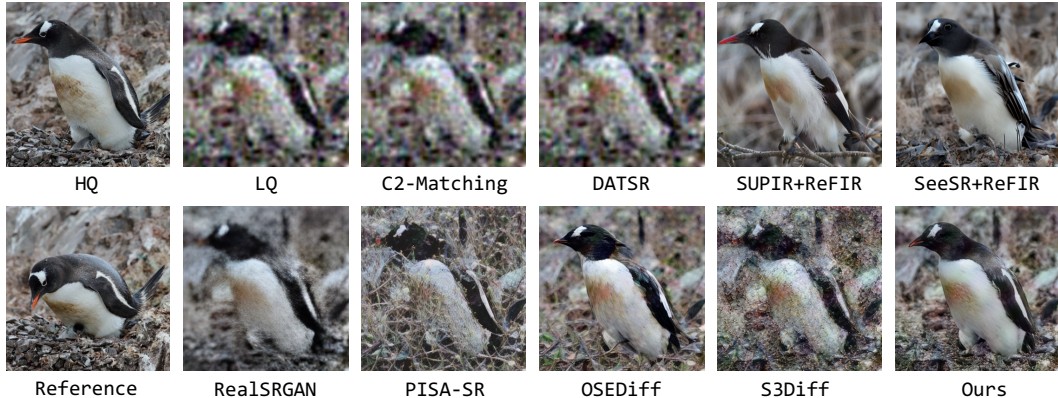

Figure 5: A Case study on WRSR dataset: To highlight differences, we cropped key regions from HQ images and their References, showing comparisons across all methods.

**Qualitative Comparison.** In Fig. 5, we compare representative approaches for scene reference generation. Traditional methods (e.g., C2-Matching, DATSR) fail under strong degradations, while single-step SR models (PISA-SR, S3Diff, OSEDiff) often produce distorted details or block artifacts. Multi-step SR with ReFIR methods alleviate some issues but still suffer from severe hallucinations—for instance, SeeSR+ReFIR reconstructs the wrong bird species and SUPIR+ReFIR generates an incorrect background—indicating limited effectiveness in reference utilization. In contrast, our method achieves a more reliable balance between fidelity and naturalness.

## 4.3 ABLATION STUDY

Table 2: Ablation study on WRSR Dataset and Bird Dataset, with the best performance highlighted in bold.

| Method | WRSR dataset | | | | Face dataset | | | |
|---|---|---|---|---|---|---|---|---|
| | PSNR ↑ | SSIM ↑ | MUSIQ ↑ | CLIPIQA ↑ | PSNR ↑ | SSIM ↑ | MUSIQ ↑ | CLIPIQA ↑ |
| **Vanilla** | 21.9508 | 0.5737 | 69.7028 | 0.6471 | 27.0795 | 0.7495 | 74.3385 | 0.6444 |
| **Global** | 21.6267 | 0.5610 | 69.0841 | 0.6281 | 27.0613 | 0.7498 | 73.1474 | 0.6308 |
| **ReFIR** | 21.7753 | 0.5668 | 68.9322 | 0.6356 | 26.9430 | 0.7473 | 74.3762 | **0.6483** |
| **AICG** | **21.9722** | **0.5777** | **69.8608** | **0.6612** | **27.1271** | **0.7523** | **74.3828** | 0.6453 |

**Full Comparison of AICG with Existing Gating Mechanisms.** To further verify the efficacy of our AICG in balancing intrinsic SR priors and reference guidance, we conduct additional ablation studies on the WRSR and Face datasets, with four method variants: (i) **Vanilla** (Vanilla RA): removes the gating term to directly use reference features; (ii) **Global** (Global learnable weight+RA): adopts the global learnable weight from (Varanka et al., 2024); (iii) **ReFIR** (ReFIR+RA): integrates ReFIR's spatial gating (Guo et al., 2024) into vanilla reference attention; (iv) **AICG** (AICG+RA): our full proposed method. All variants are trained for 11K steps under identical settings.

The results in Table 2 show that AICG consistently outperforms the Vanilla baseline across datasets, confirming the effectiveness of our design. Unlike ReFIR, which shows some gains on the Face dataset mainly in no-reference metrics but fails on dense scenarios like WRSR, AICG maintains stable improvements across both structured cases (e.g., faces with clear local correspondences) and dense scene-level references. Guided by our *"Trust but Verify"* principle, it exploits reliable cues while suppressing mismatches, achieving robust gains under diverse LQ–Ref alignment conditions.

**Visualization of AICG.** Fig. 6 shows AICG filters out misleading reference cues: irrelevant backgrounds are suppressed, and even if key structures are absent, generated regions are down-weighted. This shows AICG mitigates artifacts and regulates local details, aligning with *"Trust but Verify"* design philosophy.

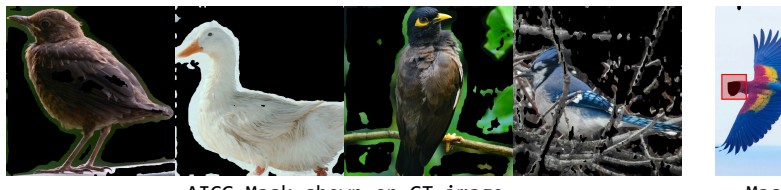

AICG Mask shown on GT image          Masked GT    Retrieved Ref

Figure 6: Visualization of our AICG mechanism (threshold 0.5). The gate suppresses irrelevant background regions while adaptively masking missing details (e.g., the bird's head), enabling fine-grained control of reference utilization.

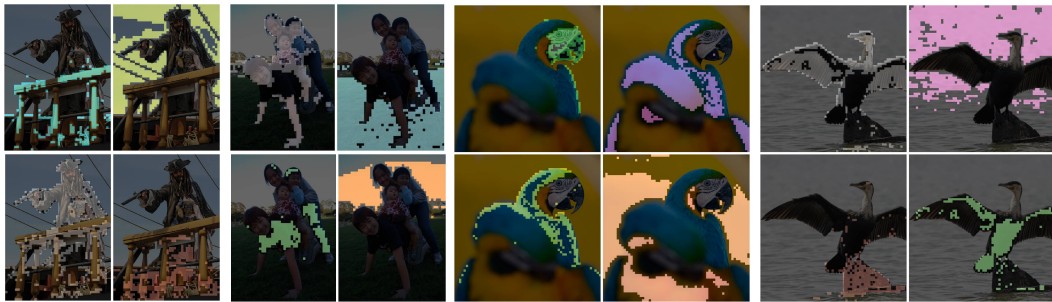

Figure 7: Visualization of learnable summary tokens. Each token captures distinct semantic regions, such as human parts, sky, grass, or bird-specific features, while others represent background context.

**Interpretability of Learnable Summary Tokens.** In our network, learnable queries act as clustering centers that aggregate reference regions into compact semantic tokens. Fig. 7 visualizes several samples, showing that tokens specialize in distinct patterns—e.g., human parts, sky, grass, or bird-specific cues like beaks and feathers—while others capture background context. These results confirm that our token design effectively condenses high-dimensional reference features into interpretable, semantically meaningful representations.

## 4.4 PERFORMANCE ANALYSIS: ROBUSTNESS AND EFFICIENCY

Table 3: GPU Memory and Inference Time at Different Resolutions.

| $512 \times 512$ | Mem (GB) | Time (s) |
| --- | --- | --- |
| S3Diff | 5.82 | 0.30 |
| Ada-RefSR | 12.66 | 0.41 |
| SeeSR+ReFIR | 11.94 | 9.03 |

| $1024 \times 1024$ | Mem (GB) | Time (s) |
| --- | --- | --- |
| S3Diff | 7.18 | 0.74 |
| Ada-RefSR | 15.54 | 1.35 |
| SeeSR+ReFIR | 18.95 | 40.23 |

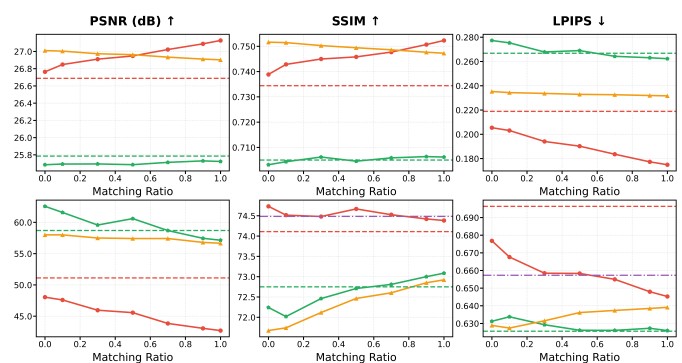

Figure 8: Comparison of different methods on the face dataset under varying reference matching ratios. Our method maintains superior reference-based metrics and naturalness even with mismatched references.

**Robustness of our method.** As shown in Fig. 8, both our method and SUPIR+ReFIR benefit from higher LQ–Ref alignment. However, SUPIR+ReFIR falls below its baseline (SUPIR) when the ratio $< 0.7$, showing sensitivity to noisy references. In contrast, our method consistently surpasses its baseline (S3Diff) even at ratio=0 and achieves no-reference scores close to HQ images,

demonstrating stronger robustness. Notably, DMDNet even degrades in PSNR/SSIM, highlighting the difficulty of maintaining fidelity under misaligned references.

**Efficiency and Model Size of Ada-RefSR.** Table 3 provides a comprehensive analysis of our model's performance footprint. Our proposed Ada-RefSR is nearly $30\times$ faster than the optimization-heavy SeeSR+ReFIR model when generating $1024 \times 1024$ images, underscoring its superior efficiency. Our core model contains $2678.89$ M parameters—about twice that of S3Diff ($1326.77$ M). Among them, only $61.98$ M parameters from reference attention and $0.20$ M from learnable summary tokens are trainable, enabling effective fine-tuning. Despite this larger capacity, the inference time compared to S3Diff only roughly doubles. This is an expected trade-off for the necessary reference processing and the enhanced personalization capability. Future work will explore lightweight injection strategies, such as token pruning or sparse attention, to further minimize computational cost.

**Efficiency of AICG.** A key design objective of AICG is to maintain minimal overhead relative to vanilla Reference Attention (RA). While ReFIR incurs a substantial cost increase of $16.0\%$ due to its explicit $L_{src} \times L_{ref}$ correlation matrix, AICG replaces this operation with an implicit estimation using only $M{=}16$ summary tokens. As confirmed by the numerical analysis in Appendix E (Table 6), this design adds merely $0.13\%$ overhead on top of RA and eliminates the need to materialize large similarity matrices, offering both computational and memory efficiency.

## 5 CONCLUSIONS

We proposed **Ada-RefSR**, a single-step reference-based super-resolution (RefSR) framework grounded in the *"Trust but Verify"* principle, addressing a key challenge in RefSR—reliably leveraging reference information under real-world degradations, where LQ–Ref mismatches often cause existing methods to overuse or misinterpret reference cues. Central to our design is the **Adaptive Implicit Correlation Gating (AICG)** mechanism, which not only implicitly models LQ–Ref correlations to generate confidence-aware gates but is also **lightweight**, introducing minimal computational overhead. This allows Ada-RefSR to exploit reliable reference cues effectively while suppressing misleading ones without sacrificing efficiency. Extensive experiments across diverse datasets confirm that Ada-RefSR achieves a strong balance between fidelity and perceptual quality, consistently outperforming state-of-the-art RefSR methods. Future directions include exploring patch-wise reference guidance for finer control and developing alternative injection mechanisms to further improve flexibility and robustness under diverse reference conditions.

## REPRODUCIBILITY STATEMENT

We provide comprehensive details to ensure reproducibility of our work. The construction of our synthetic training dataset and the bird retrieval benchmark are described in Sec. A.1 and Sec. A.2, respectively. Our training objectives, including reconstruction, perceptual, and adversarial losses, are formally defined in Sec. B. Implementation details such as network architecture, optimization settings, and training schedules are reported in Sec. section 4.1. To further support reproducibility, we submit in the supplementary materials the complete bird retrieval test dataset as well as the face test images used in our experiments. Other public datasets (DIV2K, DIV8K, Flickr2K, WRSR, CUFED5) are available online and can be accessed following the protocols described in the appendix.

## USE OF LARGE LANGUAGE MODELS (LLMS)

We used large language models (LLMs) solely for writing assistance and literature understanding (e.g., polishing style, improving clarity, summarizing related works). All ideas, methodology, experiments, and analyses were conducted by the authors, who take full responsibility for the content of this paper.

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

## Appendix Overview

The appendix is organized as follows:

## A  More details for dataset

### A.1  General Reference dataset for Training

Since no high-quality scene reference dataset exists online—most available datasets are of too low resolution to support diffusion-based SR—we construct a synthetic scene reference dataset. Specifically, we use DIV2K, DIV8K, and Flickr2K as sources, from which $1024{\times}1024$ regions are cropped with a sliding window. Each region is further split into two $512{\times}512$ sub-regions, one serving as the HQ image and the other as the Ref. To enhance robustness against distortions and better simulate real-world reference variations, the Ref images are additionally augmented via random rotations. Finally, LQ images are generated from the HQ images using the RealSRGAN degradation pipeline (Wang et al., 2021), resulting in paired LQ-HQ-Ref triplets for training.

### A.2  Bird Retrieval Dataset

To construct a high-quality retrieval dataset for evaluating reference enhancement within the same category, we collected a bird retrieval dataset consisting of 8,460 high-resolution bird images sourced from the web. We first used BLIP to filter out irrelevant images without birds, and then applied YOLOv10 (Wang et al., 2024a) to detect birds and extract the largest square crops containing them. Among these images, 66 were randomly selected as the test set, while the remaining images served as the high-quality reference database. Low-quality (LQ) images were generated from the test set using the degradation pipeline of RealSRGAN to form LQ-HQ pairs.

For retrieval, given an LQ image $\mathbf{X}_{lq}$, we searched for a semantically relevant reference $\mathbf{X}_{ref}$ from the database $\mathbb{D}$ via cosine similarity in the feature space:

$$\mathbf{X}_{ref} = \arg \max_{\mathbf{X}_{ref}^{(i)}\in\mathbb{D}} \mathrm{sim}\Big(\phi(\mathbf{X}_{lq}), \phi(\mathbf{X}_{ref}^{(i)})\Big), \tag{8}$$

where $\phi(\cdot)$ is an image feature encoder and $\mathrm{sim}(\cdot,\cdot)$ denotes cosine similarity. We adopt CLIP (Radford et al., 2021) and DINOv2 (Oquab et al., 2023) as feature extractors, as they provide robust semantic representations that suppress noise unrelated to image content.

Representative samples are shown in Fig. 9, where each sample includes the low-quality (LQ) input, its high-quality ground truth (GT), and the retrieved reference (Ref)—with similarity scores (computed via DINOv2) annotated to quantify retrieval relevance. These scores directly reflect the matching degree between LQ and Ref.

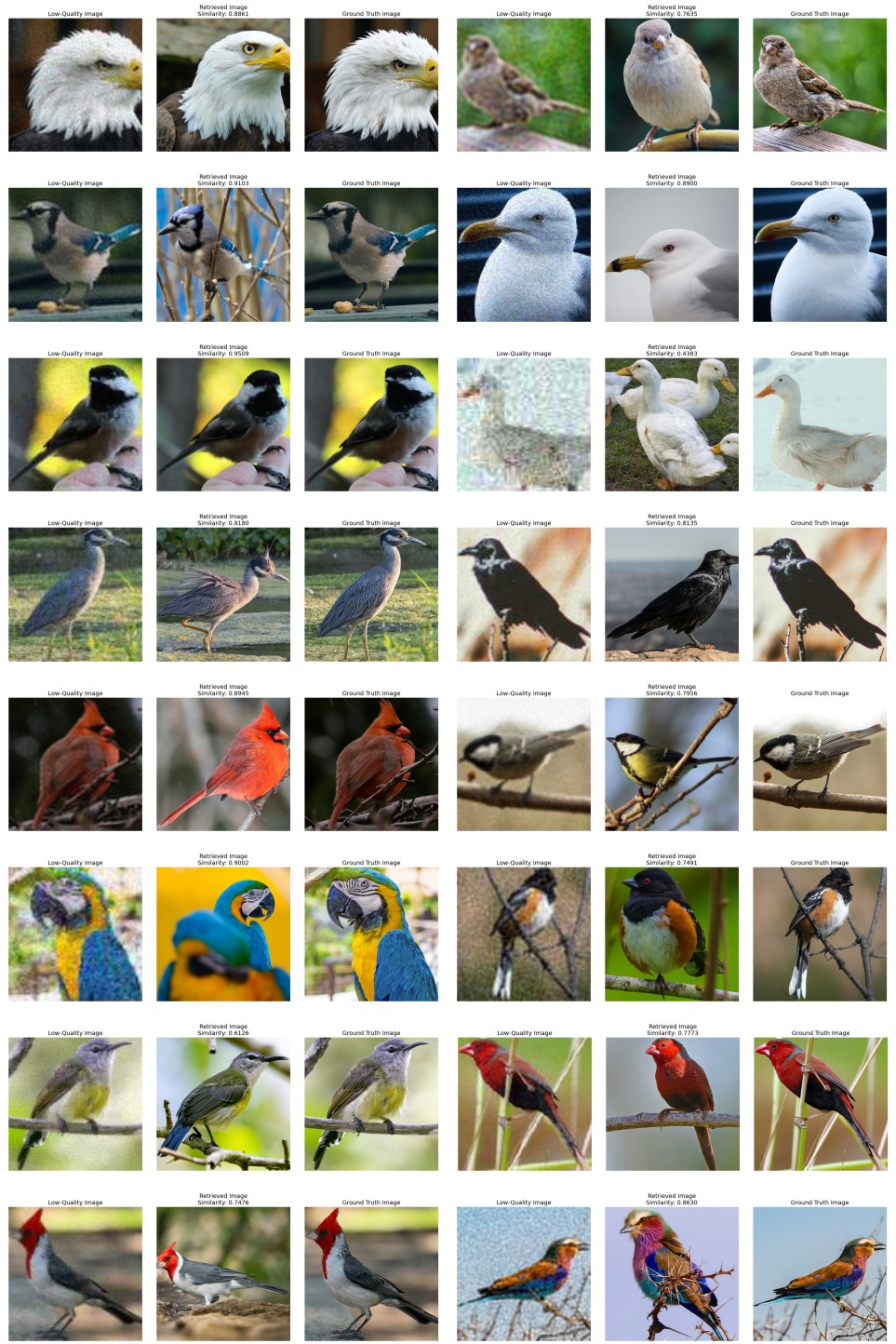

Figure 9: Representative samples from the bird retrieval dataset, including the Low quality (LQ) input, its Ground Truth image (HQ), and the retrieved reference (Ref).

# B  LOSS FUNCTIONS

Our reference based SR network is trained with a weighted combination of reconstruction, perceptual, and adversarial losses:

$$\mathcal{L}_{\text{total}} = \lambda_1 \mathcal{L}_{\text{rec}} + \lambda_2 \mathcal{L}_{\text{per}} + \lambda_3 \mathcal{L}_{\text{adv}}. \tag{9}$$

The reconstruction loss is defined as

$$\mathcal{L}_{\text{rec}} = \left\| G_\theta(\mathbf{X}_{lq}; \mathbf{X}_{ref}) - \mathbf{X}_{gt} \right\|_2^2, \tag{10}$$

the perceptual loss is defined as

$$\mathcal{L}_{\text{per}} = \frac{1}{C} \sum_{i=1}^{C} \left\| \psi_i(G_\theta(\mathbf{X}_{lq}; \mathbf{X}_{ref})) - \psi_i(\mathbf{X}_{gt}) \right\|_2^2, \tag{11}$$

where $\psi_i(\cdot)$ denotes the $i$-th feature map from a pretrained VGG backbone (Simonyan & Zisserman, 2014). The adversarial loss adopts the standard GAN formulation:

$$\begin{aligned} \mathcal{L}_{\text{adv}} =& \mathbb{E}_{\mathbf{X}_{gt} \sim p_{\text{data}}} \left[ \log D_\omega(\mathbf{X}_{gt}) \right] \\ &+ \mathbb{E}_{(\mathbf{X}_{lq}, \mathbf{X}_{ref}) \sim p_{\text{data}}} \left[ \log \left( 1 - D_\omega(G_\theta(\mathbf{X}_{lq}, \mathbf{X}_{ref})) \right) \right] \end{aligned} \tag{12}$$

where $G_\theta$ is our Ada-RefSR model and $D_\omega$ is the discriminator (Caron et al., 2021).

# C  VALIDATION FOR REFERENCE ATTENTION BASED FEATURE INJECT

## C.1  RELATIVE POWER SPECTRUM VISUALIZATION

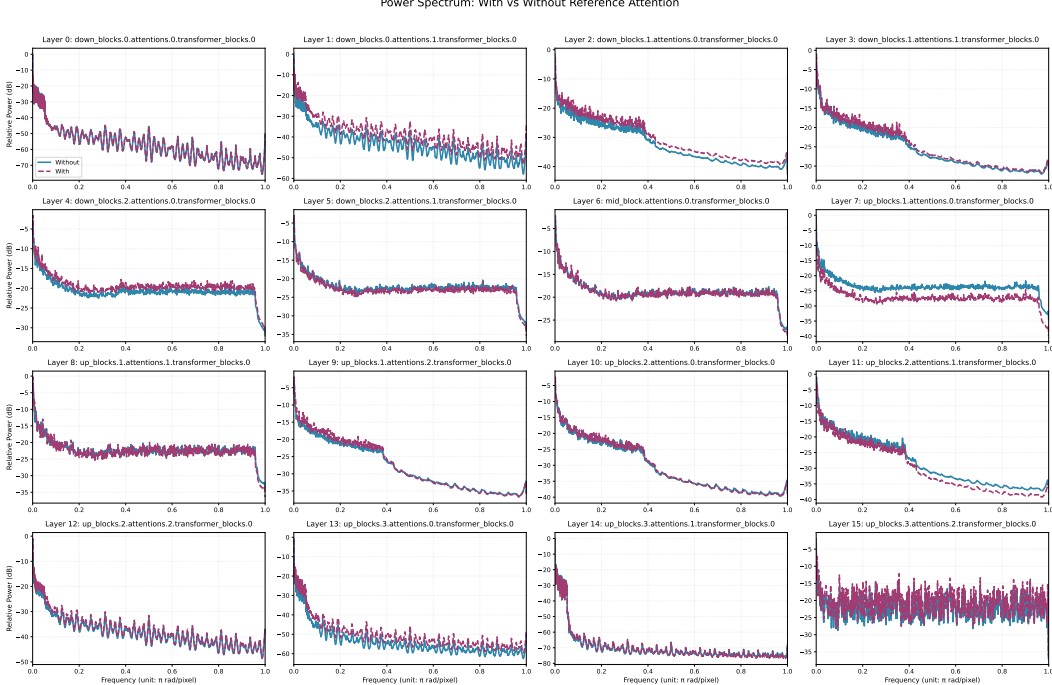

Figure 10: Visualization of the Relative Power Spectrum for Intermediate Feature Maps.

We visualize the power spectrum response curves on the CUFED5 dataset to validate the effectiveness of our reference attention in injecting reference information from a frequency-domain perspective. Specifically, we compare the output results of attn1 (first attention module) across all

Transformer Blocks—with a total of 16 layers available for comparison. Here, "Without" denotes the intermediate-layer outputs of the original baseline model, while "With" represents the frequency responses after integrating our reference attention mechanism.

As shown in the Fig. 10, after introducing reference attention, the frequency responses are improved across most layers, with notable enhancements in the top layers (Layer 1, 2, 13). This indicates that reference attention enables the model to more completely and balancedly retain and enhance the local detail information of features. It further confirms that the introduction of reference images primarily achieves the transfer of image details and textures, ultimately leading to clearer visual details and stronger realism in the generated results.

# D  HOW MANY SUMMARY TOKENS ARE NEEDED?

Table 4: Comparison of token numbers on WRSR and Bird datasets across multiple metrics.

| Dataset | Token | PSNR | SSIM | LPIPS | FID | NIQE | MUSIQ | CLIPIQA |
|---------|-------|------|------|-------|-----|------|-------|---------|
| **WRSR** | 8 | 21.8914 | 0.5716 | 0.3063 | 53.1421 | **3.6916** | 69.4745 | 0.6439 |
| | 16 | **21.9722** | **0.5777** | 0.3061 | 53.2811 | 3.7429 | **69.8608** | **0.6612** |
| | 32 | 21.9127 | 0.5764 | **0.3055** | **52.5586** | 3.7712 | 69.7996 | 0.6569 |
| **Bird** | 8 | 25.2595 | 0.7263 | 0.2556 | 36.5373 | **5.0629** | 71.3751 | 0.8216 |
| | 16 | **25.2998** | **0.7292** | **0.2536** | **36.4249** | 5.1771 | 72.0039 | **0.8303** |
| | 32 | 25.2286 | 0.7273 | 0.2538 | 38.2489 | 5.1859 | **72.1367** | 0.8234 |

Herein, we further conduct experiments on the required number of learnable tokens. We set the number of learnable tokens to 8, 16, and 32 respectively, with detailed results reported in Table 4.

As shown in Table 4, when the number of learnable tokens is set to 16, the better performance is achieved on both the CUFED5 and Bird datasets. This indicates that learnable tokens do not need to be excessive, and a small number of tokens can already exert their effectiveness.

# E  COMPUTATIONAL COMPLEXITY ANALYSIS

In this section, we provide a theoretical comparison between the computational costs of **Explicit Correlation-based Gating** in ReFIR (Method 1, denoted as $m_1$) and our proposed **Implicit Correlation-based Gating** (Method 2, denoted as $m_2$).

**Notation.** We analyze the additional computation introduced by each gating mechanism relative to the shared Reference-Attention (RA) module. We define the following dimensions:

- $\mathbf{H}_{src} \in \mathbb{R}^{L_{src} \times d}$: Source feature sequence.
- $\mathbf{H}_{ref} \in \mathbb{R}^{L_{ref} \times d}$: Reference feature sequence.
- $\mathbf{T}_S \in \mathbb{R}^{M \times d}$: Learnable summary tokens (specific to Method 2).

## E.1  COST DERIVATION

**Reference Attention Baseline.** Both methods share the underlying RA computation. This involves five linear projections (to_q, to_k, to_v, zero_linear, to_out). The base computational cost is:

$$\mathcal{C}_{\text{base}} = (3L_{src} + 2L_{ref})d^2 + 4L_{src}L_{ref}d. \tag{13}$$

**Method 1: Explicit Correlation.** Method 1 calculates the full token-to-token cosine similarity matrix between $\mathbf{H}_{src}$ and $\mathbf{H}_{ref}$. This introduces an explicit interaction cost of $2L_{src}L_{ref}d$. The total cost is:

$$\begin{aligned}
\mathcal{C}(m_1) &= \mathcal{C}_{\text{base}} + 2L_{src}L_{ref}d \\
&= (3L_{src} + 2L_{ref})d^2 + 6L_{src}L_{ref}d. \tag{14}
\end{aligned}$$

**Method 2: Implicit Correlation (Ours).** Method 2 avoids full pair-wise similarity by introducing two lightweight steps:

1. **Reference Summarization:** Summary tokens attend to reference features (reusing RA projections), costing $Md + 2L_{ref}Md$.
2. **Implicit Estimation:** Source queries interact only with the summary tokens $\mathbf{T}_S$, costing $2L_{src}Md$.

Summing these with the baseline, the total cost is:

$$\begin{aligned}\mathcal{C}(m_2) &= \mathcal{C}_{\text{base}} + Md + (2L_{ref} + 2L_{src})Md \\ &= (3L_{src} + 2L_{ref})d^2 + 4L_{src}L_{ref}d + Md + (2L_{ref} + 2L_{src})Md.\end{aligned} \quad (15)$$

### E.2 ASYMPTOTIC COMPARISON

To clearly visualize the efficiency gain, we consider the typical setting where sequence lengths dominate the summary token count and hidden dimension:

$$L_{src} = L_{ref} = L, \quad L \gg M, \quad L \gg d.$$

Under these conditions, the dominant complexity terms simplify as shown in Table 5.

Table 5: Complexity comparison of gating mechanisms. Method 2 significantly reduces the coefficient of the quadratic term.

| Method | Dominant Term | Interaction Type | Complexity |
|---|---|---|---|
| Explicit ($m_1$) | $6L^2d$ | All-to-All | $\mathcal{O}(L^2)$ |
| **Implicit** ($m_2$) | $4L^2d$ | Summarized | $\mathcal{O}(L^2)$ |

The relative computational load is approximately:

$$\frac{\mathcal{C}(m_2)}{\mathcal{C}(m_1)} \approx \frac{4L^2d}{6L^2d} \approx 0.67. \quad (16)$$

Thus, the proposed implicit correlation gating reduces the attention complexity burden by roughly **33%** while maintaining robust adaptivity.

### E.3 NUMERICAL EVALUATION

To quantify the practical overhead introduced by each method, we substitute typical high-resolution settings used in our experiments. We assume an input image size of $512 \times 512$, resulting in a sequence length $L = 4096$ ($64 \times 64$ latent space after $8\times$ VAE downsampling), along with a hidden dimension $d = 1024$, and $M = 16$ summary tokens.

**Overhead Analysis.** Based on Eq. equation 13, the baseline Reference Attention (RA) requires approximately $2.15 \times 10^{11}$ FLOPs.

- **Method 1 (Explicit):** Calculating the full $L \times L$ similarity matrix introduces a substantial overhead of $\approx 3.44 \times 10^{10}$ operations, increasing the total computational cost by **16.0%**.
- **Method 2 (Implicit, Ours):** By leveraging $M = 16$ summary tokens, the additional cost is drastically reduced to $\approx 2.69 \times 10^8$ operations. This represents a negligible overhead of only **0.13%** over the baseline.

As shown in Table 6, while Explicit Gating imposes a heavy computational burden, our Implicit Gating achieves adaptivity with virtually zero additional cost ($\sim 127\times$ more efficient in terms of added overhead).

Table 6: Numerical complexity comparison ($L = 4096, d = 1024, M = 16$). Our Method 2 introduces negligible overhead compared to the baseline, whereas Method 1 adds significant cost.

| Module | Total FLOPs | Added Cost | Relative Overhead |
|---|---|---|---|
| Baseline RA | $2.15 \times 10^{11}$ | – | – |
| Explicit Gating ($m_1$) | $2.50 \times 10^{11}$ | $3.44 \times 10^{10}$ | $+16.00\%$ |
| **Implicit Gating ($m_2$)** | $\mathbf{2.15 \times 10^{11}}$ | $\mathbf{2.69 \times 10^8}$ | $\mathbf{+0.13\%}$ |

## F    ROBUSTNESS TO VARIOUS REFERENCE QUALITY AND MISALIGNMENT

We conducted controlled experiments on the face reference super-resolution benchmark proposed in our paper to systematically investigate the robustness of Ada-RefSR. The study considered varied reference conditions, including degraded reference quality and potential misalignment. Since all faces in this dataset are frontal, we implemented two types of synthetic degradations applied to the reference images:

1. **Gaussian Blur Degradation:** Reference images were convolved with Gaussian kernels of progressively increasing sizes (3, 5, 7, 9, and 11 pixels) to simulate varying levels of blur.

2. **Affine Degradation:** Graded affine transformations were applied to simulate potential misalignment. This included simultaneous variations in rotation angle, scale deviation, translation ratio, and corner offsets. Five intensity levels were defined, with each level increasing all transformation parameters relative to the previous, thereby quantifying the severity of misalignment.

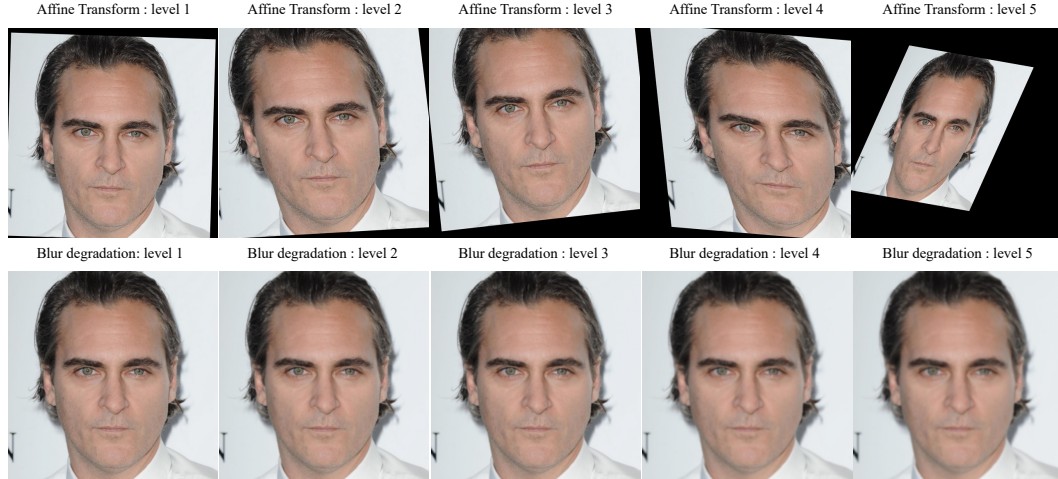

Figure 11: Examples of Gaussian blur and affine degradations on reference images at different intensity levels: affine transformations (1st row) and Gaussian blur with increasing kernel sizes (2nd row).

Figure 11 visualizes the impact of each degradation type at different intensity levels. We evaluated all reference-based face SR methods under these controlled degradations and summarized performance metrics in line charts (Fig. 12).

**Observations.**    Under affine degradation, Ada-RefSR demonstrates strong robustness: metrics such as FID and LPIPS remain largely stable even as misalignment increases, whereas alternative methods exhibit notable performance drops, highlighting their sensitivity to spatial inconsistencies. Under Gaussian blur degradation, our method maintains competitive performance relative to face-specific SR approaches such as InstantRestore, DMDNet, and FaceMe. Across nearly all metrics, Ada-RefSR consistently outperforms these baselines, even at higher degradation levels.

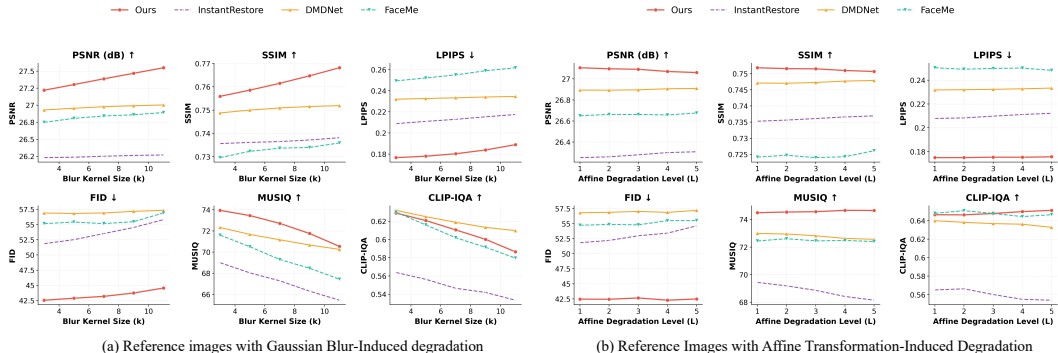

Figure 12: **Performance curves for various face RefSR methods** – Degradation level of the reference image is varied. Results are shown for two degradation types: (a) Gaussian blur-Induced degradation and (b) Affine Transformation-Induced degradation.

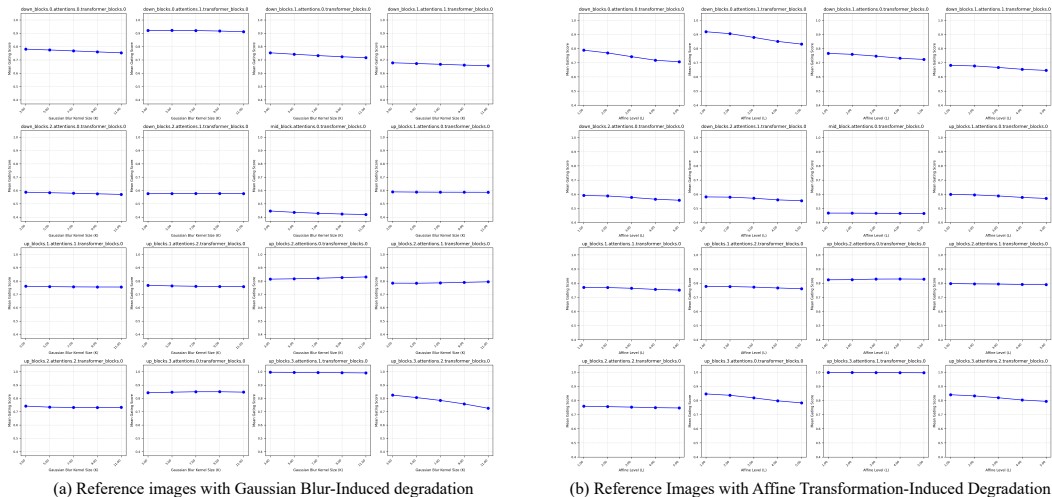

Figure 13: Average AICG gating activations across layers of the reference attention module under varying reference quality. Lower-quality references lead to reduced gating activations, illustrating adaptive down-weighting of unreliable regions.

These results indicate that Ada-RefSR can effectively leverage reference information while remaining resilient to variations in quality and alignment. This robustness is primarily attributed to the **Adaptive Implicit Correlation Gating (AICG)** mechanism.

**Interpretability Analysis.** To provide further insights, we examined the internal gating activations of AICG across layers of the reference attention module. As shown in Fig. 13, the average gating activations decrease progressively as reference quality declines. This demonstrates that AICG adaptively attenuates contributions from unreliable reference regions, thereby maximizing the utility of informative reference features while mitigating the adverse effects of poor-quality or misaligned references.

**Visualization Examples.** Figure 14 provides additional qualitative examples under different reference degradations. When the reference undergoes affine transformations, the restored faces remain highly consistent with those obtained using the clean reference, indicating strong robustness to misalignment and viewpoint inconsistencies. Under Gaussian blur, our method still preserves identity while only losing fine-grained details that are absent from the degraded reference. When an unrelated reference image is provided, our results naturally revert to those of the single-image baseline (S3Diff), confirming that the proposed AICG mechanism effectively suppresses irrelevant reference cues and prevents quality degradation. These observations further validate the "trust but verify" behavior of Ada-RefSR across diverse reference conditions.

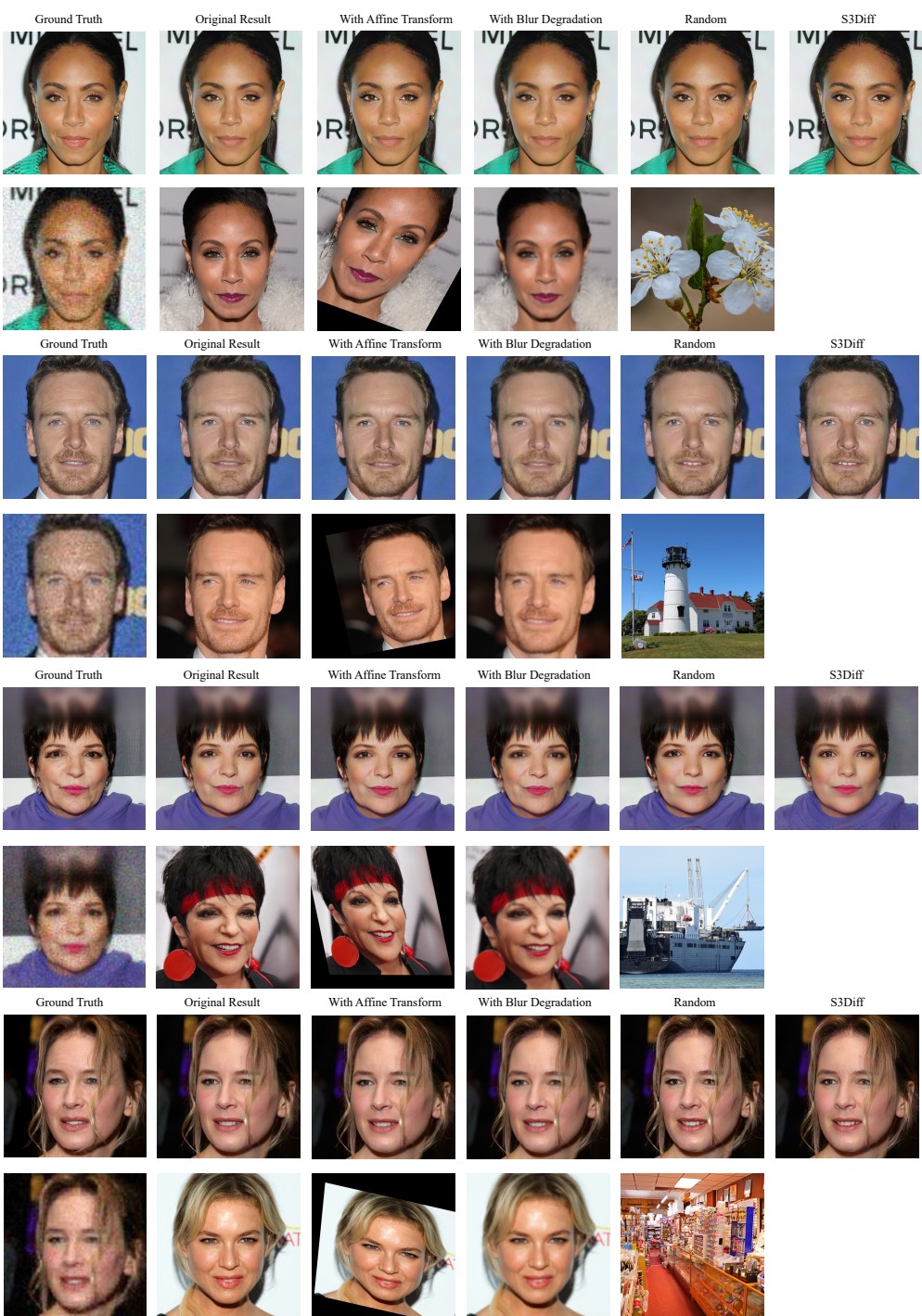

Figure 14: Generated results under different reference degradations (first row) with corresponding LQ and reference images (second row). The last column shows the S3Diff baseline. Our method remains effective under affine and blur degradations, and defaults to the baseline when using random references.

# G    MORE RESULTS ON ARTIFICIAL TEXTURES

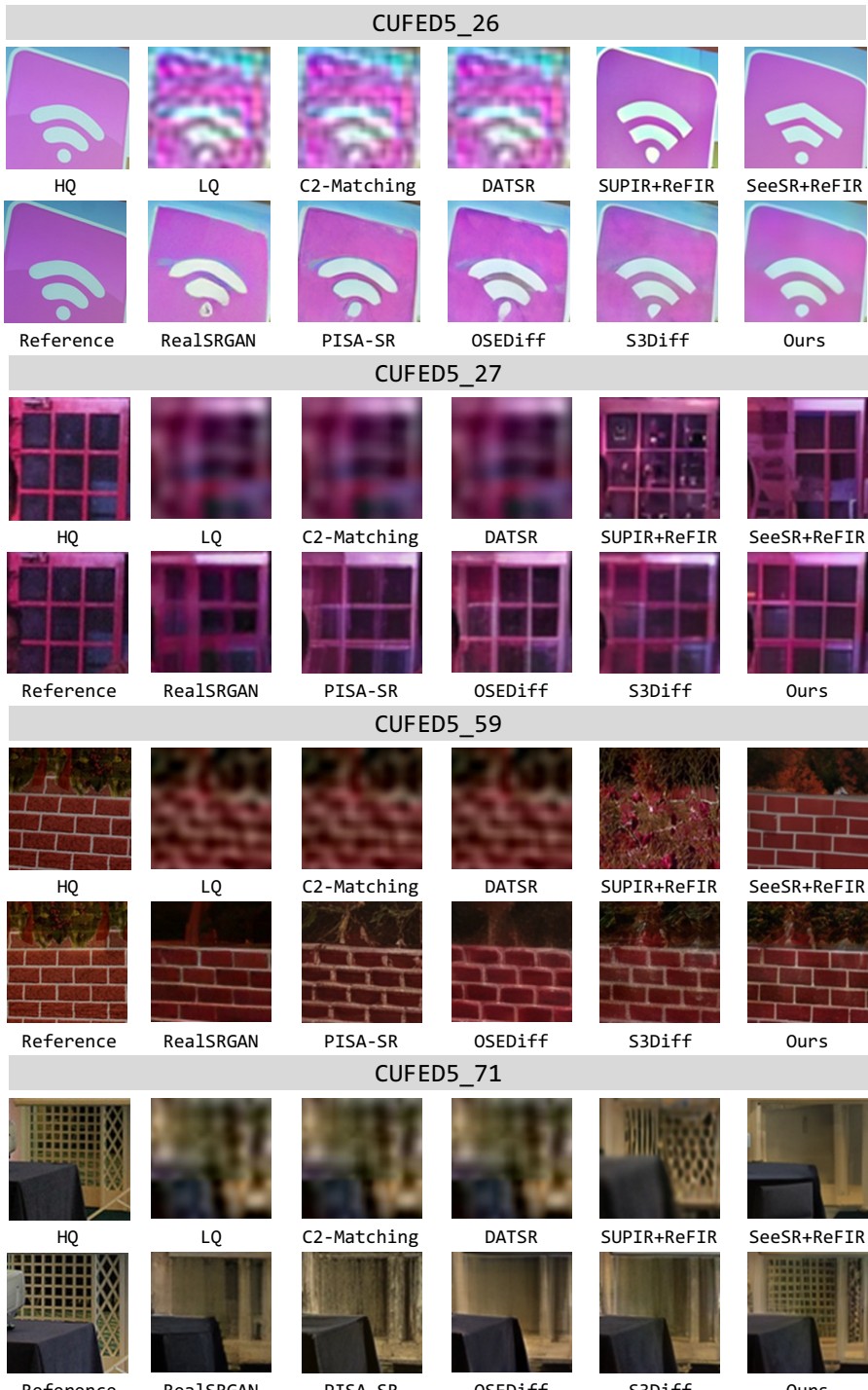

Figure 15: Visual comparisons on complex artificial textures from the CUFED5 dataset. Our method better restores contours and material textures, providing richer details.

To further demonstrate the generalization capability of our model beyond natural scenes, we provide additional visual examples involving complex artificial textures in Fig. 15 and Fig. 16. These cases include a variety of man-made patterns such as logos, wall tiles, door frames, and fabric materials.

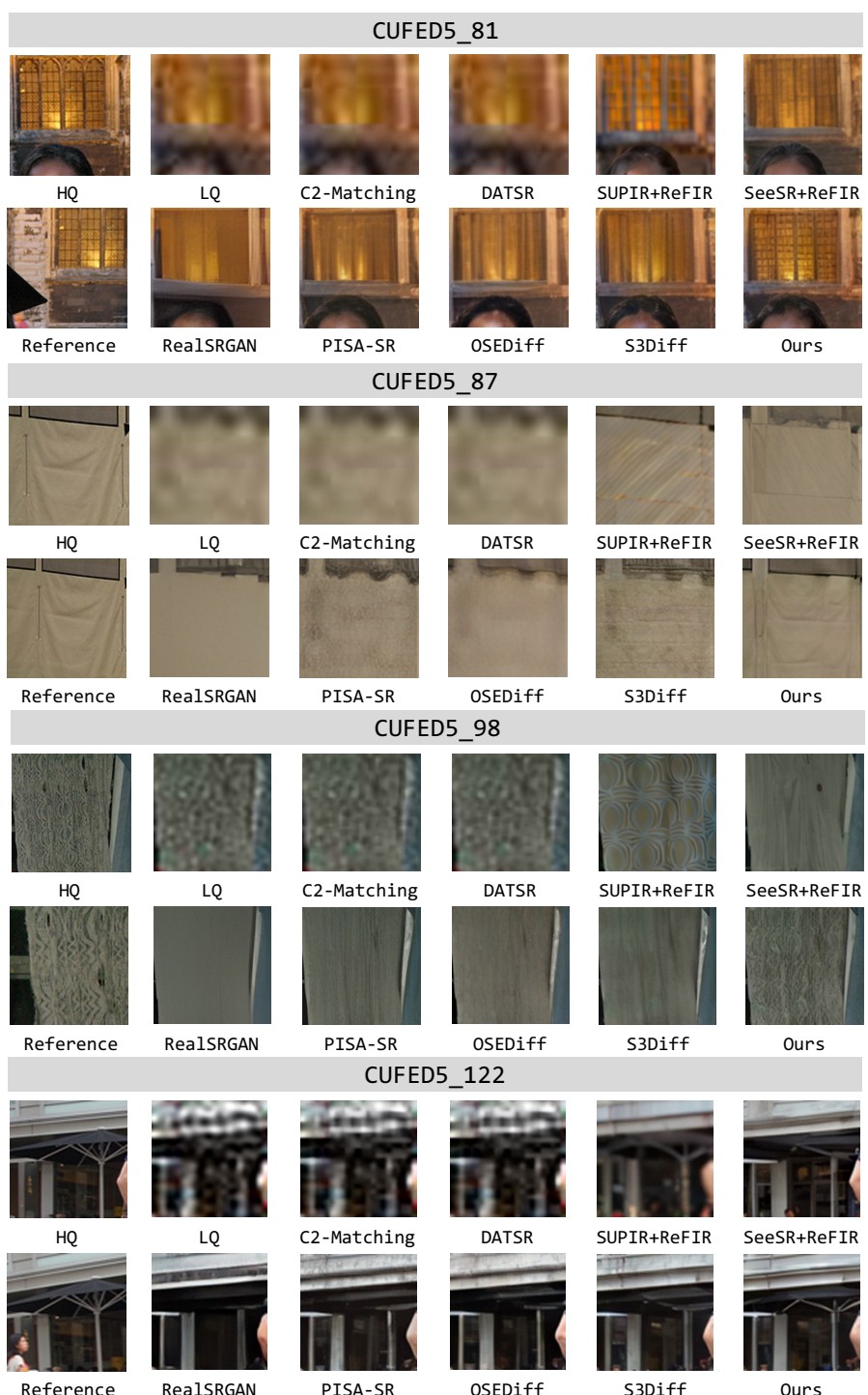

Figure 16: Visual comparisons on complex artificial textures from the CUFED5 dataset. Our method better restores contours and material textures, providing richer details.

For example, in the CUFED5_26 case (Fig. 15), both C2-Matching and DATSR fail to restore the degraded input, as they rely on the assumption of synthetic $\times 4$ downsampling and lack the ability to handle real-world degradations. Single-step diffusion SR baselines (PISA-SR, OSEDiff, S3Diff) either retain noticeable noise or produce overly smoothed textures. Although SUPIR+ReFIR and SeeSR+ReFIR are able to generate a complete WiFi logo, they exhibit notable structural inconsistencies: SUPIR+ReFIR produces unnaturally sharp boundaries and over-bright colors inconsistent with the reference, while SeeSR+ReFIR introduces a sharp and unrealistic transition near the top of the symbol. In contrast, our method not only removes the degradation effectively but also preserves both the soft boundary and the original color distribution of the logo, faithfully matching the reference.

Beyond logos, our approach also successfully recovers fine-grained man-made textures with high geometric fidelity. In CUFED5_59 (Fig. 15), our method accurately reconstructs the layout, boundary structure, and material appearance of wall tiles. In CUFED5_87 (Fig. 16), Ada-RefSR restores the fabric texture and the thin black frame in the upper region, which competing methods fail to preserve. Similarly, in CUFED5_98 (Fig. 16), our model effectively captures the curtain material and restores its structural regularity.

These examples collectively demonstrate that our model generalizes well to structured, artificial patterns. The combination of (1) a strong single-image SR prior and (2) our trust-but-verify reference integration mechanism allows Ada-RefSR to selectively leverage reference cues while maintaining geometric accuracy and material consistency across a wide range of real-world textures.

## H  MORE VISUAL RESULTS

In this section, we provide additional visual comparisons across the three evaluated datasets to further demonstrate the effectiveness of our proposed method. Specifically, the visual results for the bird retrieval, face, and WRSR datasets are illustrated in Fig. 17, Fig. 18, and Fig. 19, respectively.

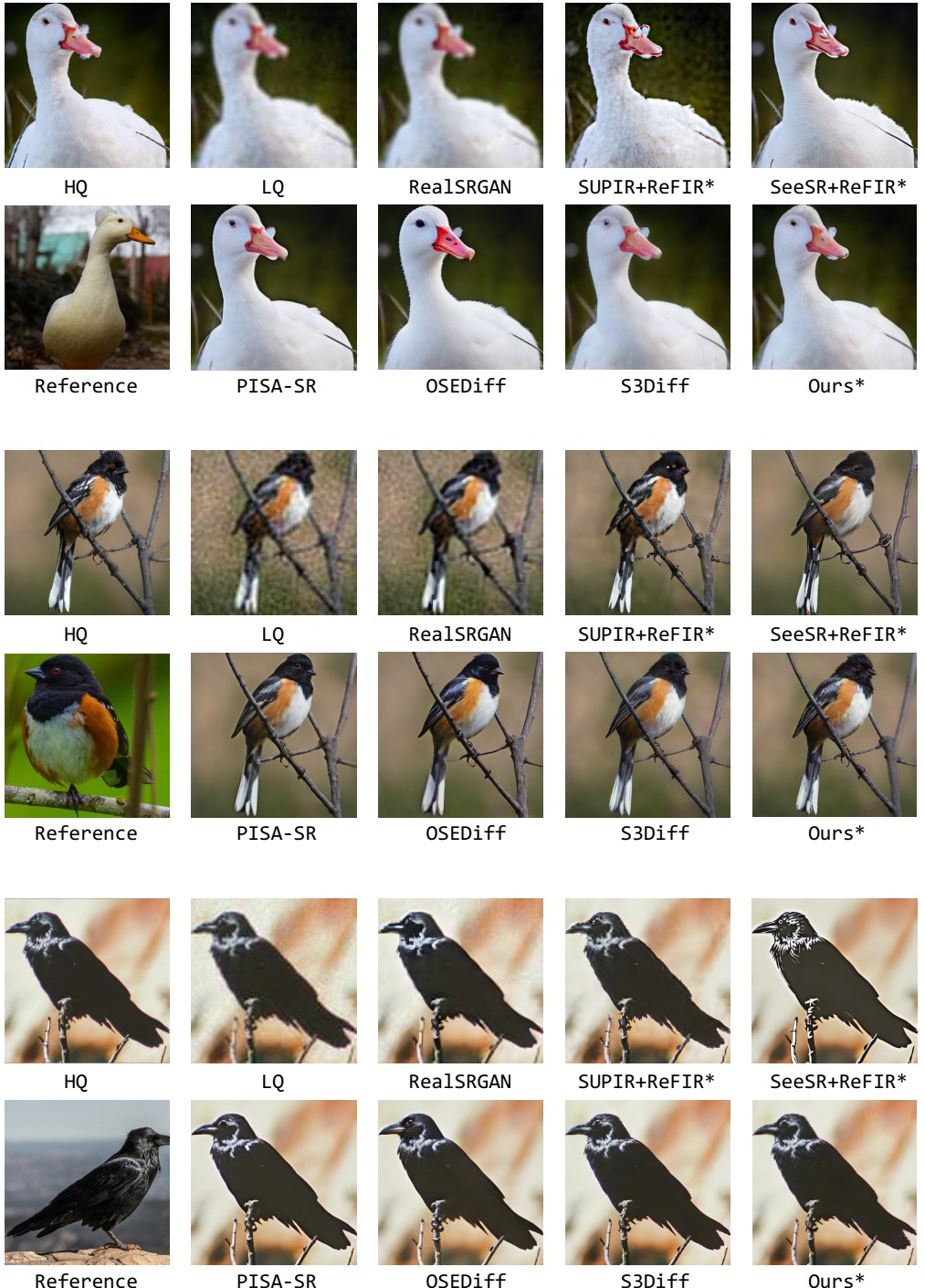

Figure 17: Visual comparison results on the bird retrieval dataset.

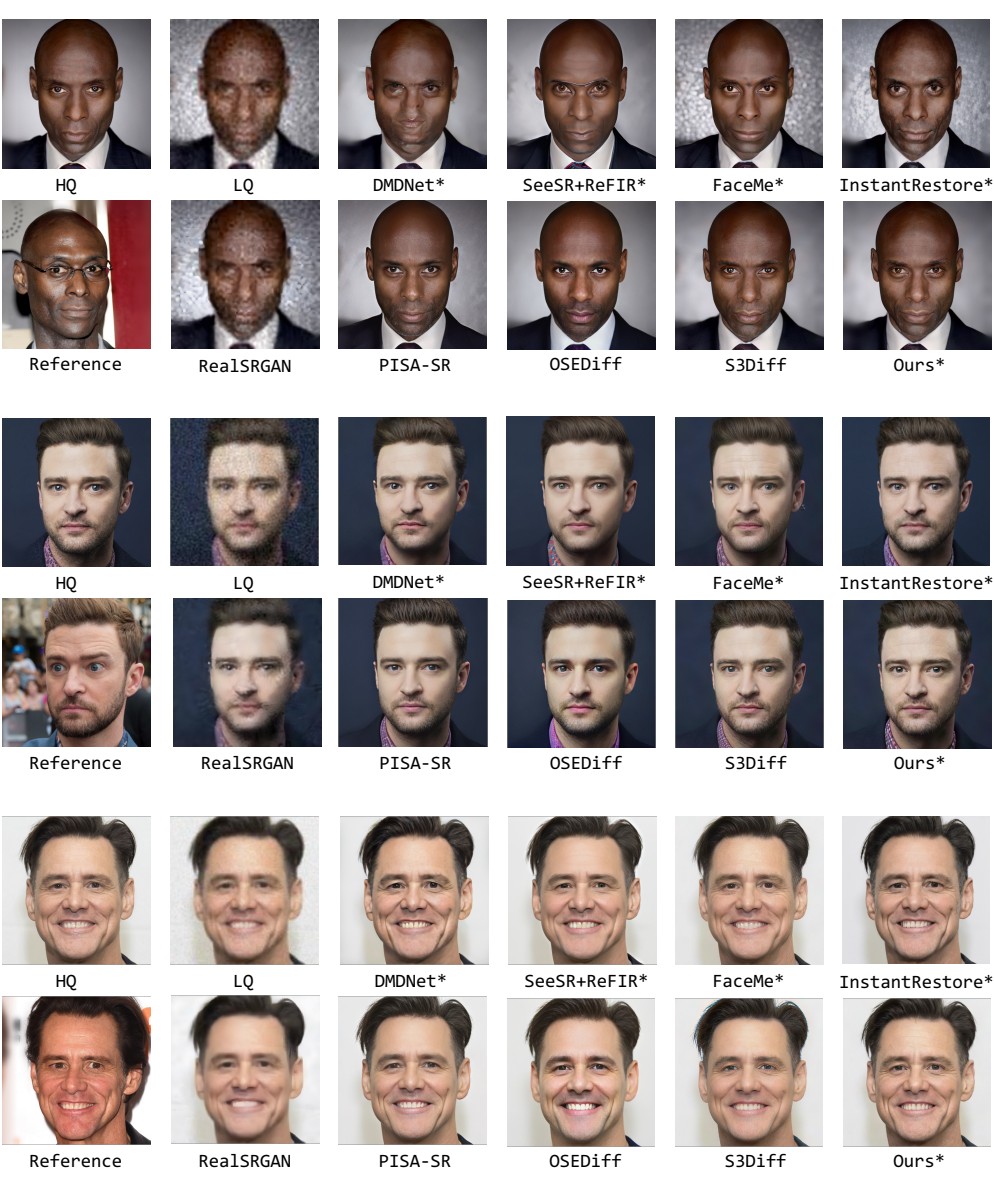

Figure 18: Visual comparison results on the face dataset.

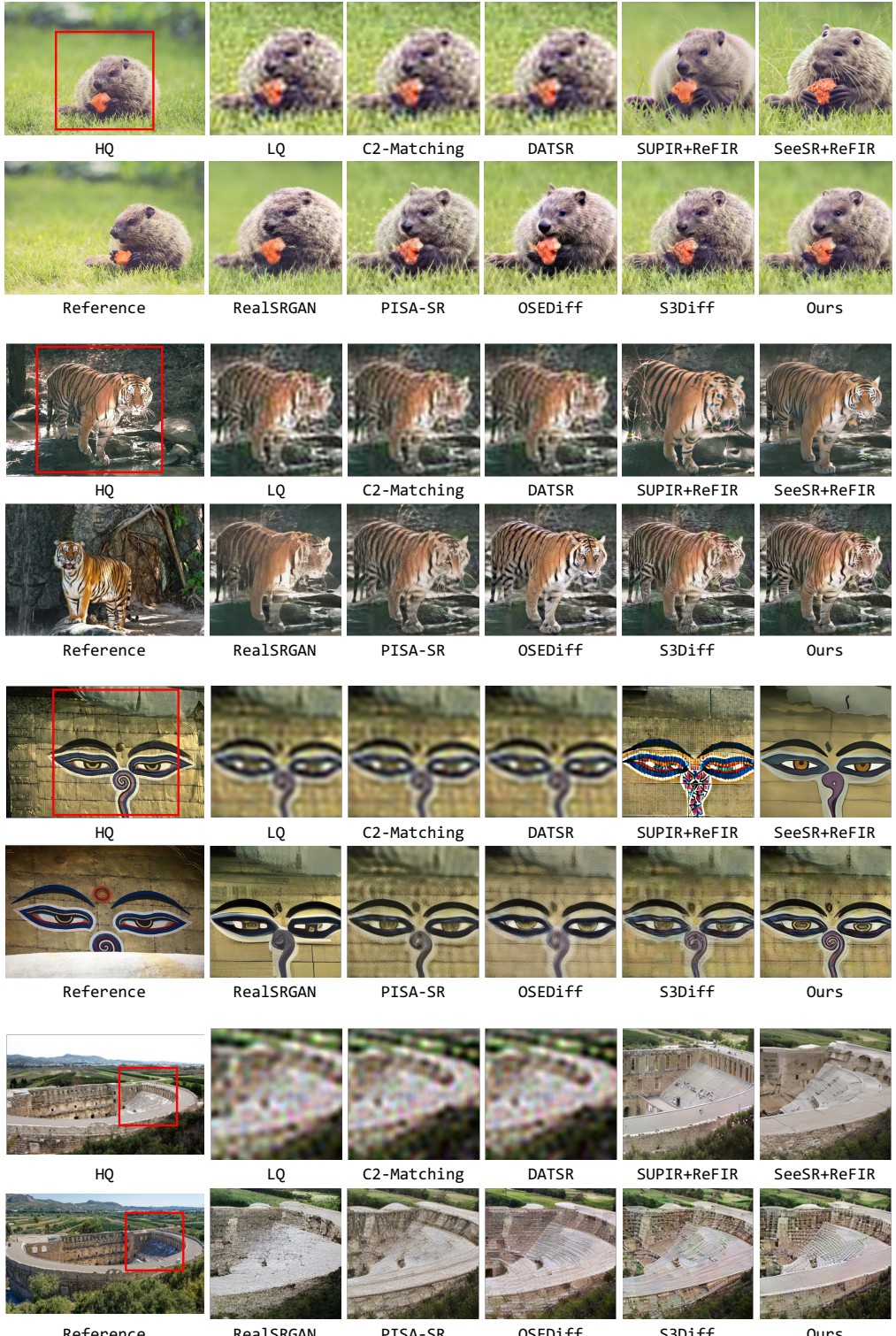

Figure 19: Visual comparison results on the WRSR dataset.

