# OpenReview forum: "Trust but Verify: Adaptive Conditioning for Reference-Based Diffusion Super-Resolution via Implicit Reference Correlation Modeling"
_ICLR.cc/2026/Conference — ICLR 2026 Poster_

### Official Review · Reviewer_Fy7H · 2025-10-30

**Soundness:** 2
**Presentation:** 3
**Contribution:** 2
**Rating:** 4
**Confidence:** 3

**Summary:**

This paper proposes Ada-RefSR, a diffusion-based model for Reference-based image super-resolution. The core idea of this paper lies in two terms: the fidelity and generalization. It proposes AICG to adaptively balance intrinsic SR fidelity with reference-guided enhancement, mitigating hallucinations. AICG enables Ada-RefSR to generalize RefSR beyond narrow domains to diverse scenarios, maintaining robustness under varying reference alignment. Further, built on one-step diffusion model, its inference is efficient compared to other multi-step diffusion model.

**Strengths:**

1.	The paper proposes AICG designed for RefSR task. Ada-RefSR injects LQ features directly using a residual connection besides the reference feature selection. It allows the model to preserve the prior knowledge. Also, a gating mechanism is applied in the reference feature attention components to adaptively select useful information from the reference image.
2.	The model achieves SOTA performance with efficient one-step diffusion model.
3.	The idea is straightforward and easy to follow.

**Weaknesses:**

1.	The novelty is limited, since most of the model design and idea like LQ feature residual connection is straightforward and has been proposed in classic non-diffusion methods. The paper should discuss the novelty specifically for Ref-SR task.
2.	Though claiming efficiency as one of the contributions, the paper provides insufficient discussions and experiments on efficiency and model size.
3.	The visualizations are mostly derived from animal images, are there results and visual comparisons on more complex textures like text or some artificial objects to better show the generalization of this model?

**Questions:**

Refering to the weaknesses. Clarify the contributions specfically for Ref-SR. The visualization of some complex textures are also important to reveal the generalization of the proposed method.

---

> ### Author Response · Authors · 2025-11-21
> **Overall Response to Reviewer Fy7H**
>
> We sincerely thank you for the constructive and insightful assessment of our work. We recognize the importance of the raised concerns regarding novelty, efficiency discussion, and visual generalization, and have addressed them with substantial revisions, new analysis, and additional experiments in the revised manuscript.
>
> ### Summary of Reviewer's Core Problems:
>
> * **Novelty:** Novelty seems limited to standard components; clarification needed on contributions specifically for the RefSR task.
> * **Efficiency & Model Size:** Insufficient analysis on computational overhead and model size compared to speedup claims.
> * **Generalization Visuals:** Lack of visual comparisons on complex textures (e.g., text, artificial objects) to verify generalization.
>
> ### Our Response and Revisions:
>
> 1.  **On Novelty:** We clarify that the novelty lies not in standard components, but in solving a key, unresolved **Real-World RefSR challenge** of reliably using reference information under real world degradation. We propose **AICG**, a novel implicit, correlation-driven gating mechanism (distinct from prior static/explicit methods). We have **restructured the Abstract, Introduction and Conclusion** to explicitly focus the contributions on this mechanism (W1 response).
>
> 2.  **On Efficiency and Model Size:** We confirm Ada-RefSR's efficiency stems from the **one-step diffusion backbone** ($\sim 30\times$ faster than multi-step baselines) and the **lightweight AICG module** (adding only 0.13% overhead vs. $\approx $16% for explicit gating). We added **Appendix E** with detailed FLOPs analysis and updated **Section 4.4** to clarify model size relative to superior runtime performance (W2 response).
>
> 3.  **On Generalization Visuals:** We have added extensive new visual comparisons on **complex, man-made textures** (WiFi logos, tiles, fabric patterns) in **Appendix G (Figures 15 and 16)**. These results empirically confirm Ada-RefSR's strong generalization and its ability to maintain structural fidelity in highly challenging structured scenes (W3 response).
>
> We believe these revisions and added evidence comprehensively address all concerns, substantially strengthening our claims regarding novelty, robustness, efficiency, and generalization.

---

> ### Author Response · Authors · 2025-11-21
> **Response to weaknesses 1**
>
> We appreciate your concern. We clarify that the novelty of our work does *not* lie in residual connections or other standard architectural components, but in addressing a key and unresolved challenge in reference-based super-resolution (RefSR): **how to reliably use reference information under real-world degradations**.
>
> ### 1. The Unsolved Problem in Real-World RefSR
>
> Classical non-diffusion RefSR methods assume synthetic $\times 4$ downsampling, where LQ–Ref correspondence is clean and stable. In real-world RefSR, however:
>
> * The LQ input suffers from unknown degradations (blur, noise, compression).
> * The spatial correspondence between LQ and Ref becomes unreliable, making **explicit matching**—the foundation of most prior RefSR pipelines—highly unstable.
>
> This necessitates a robust **gating mechanism**: the model must decide *when to trust* the reference and *when to suppress it* under misalignment or low-quality references. Existing designs struggle here: PFStorer uses a fixed global gate unrelated to alignment quality, while ReFIR relies on explicit token correlations that are extremely sensitive to noise and long-tailed similarity statistics.
>
> ### 2. Our Core Novelty: The AICG Mechanism
>
> We propose a new gating mechanism, **AICG**, explicitly designed to address these RefSR-specific challenges:
>
> * **Implicit correlation modeling:** Instead of computing raw LQ–Ref similarity (ReFIR) or relying on a static gate (PFStorer), AICG introduces learnable summary tokens that implicitly encode cross-image correlations in a noise-tolerant manner. To our knowledge, this is the first RefSR gating strategy that is both correlation-driven and free from explicit matching.
> * **Degradation-aware reference reliability estimation:** AICG predicts **token-wise** reliability scores conditioned jointly on LQ content and alignment quality, enabling the model to down-weight ambiguous or misleading reference regions—a widely observed failure mode of existing RefSR approaches.
> * **Lightweight diffusion-oriented design:** AICG reuses RA projections and intermediate computations, introducing only a handful of learnable summary tokens. This achieves **negligible additional FLOPs** while offering a more stable and adaptive reference-injection pathway than explicit-correlation baselines.
>
> ### 3. Significance and Contribution
>
> Although both diffusion SR and RefSR are active research areas, *no prior work has introduced an implicit, adaptive, and computation-efficient gating mechanism tailored for real-world RefSR*. Our approach offers a new "**trust but verify**" perspective and provides a practical solution that:
>
> * Improves robustness under misalignment, degradation, and noisy references.
> * Avoids the brittleness of explicit matching.
> * Generalizes across faces, dense scenes, and class-level references.
>
> **In summary:** The novelty does not stem from residual paths, but from directly addressing the fundamental bottleneck of real-world RefSR through a new implicit, correlation-driven gating mechanism that is robust, adaptive, and lightweight.

---

> ### Author Response · Authors · 2025-11-21
> **Revision in the manuscript for weakness 1**
>
> **Revision in the manuscript.**
> To ensure the manuscript fully reflects the methodological novelty discussed in the rebuttal, we have made the following updates to the Abstract and the Introduction's Contributions section.
>
> ---
>
> ### Abstract Refinement
>
> The abstract has been revised to better highlight the **novelty and practical significance** of our method, specifically integrating the failure of **explicit matching under real-world degradations** as the central problem.
>
> * The background is clarified to emphasize the challenge of unreliable LQ–Ref correspondences in real-world degradations and the resulting importance of adaptive gating.
> * The description of **AICG** has been updated to explicitly note its **lightweight, adaptive** nature, highlighting its efficiency in regulating reference guidance while preventing erroneous fusion.
>
> ---
>
> ### Introduction Contribution Reframing (Focus Shift)
>
> The list of contributions at the end of the Introduction has been reorganized to directly capture the **Mechanism** and the **Systemic Impact**.
>
> #### Original Contributions:
>
> > * ***Fidelity.*** *We propose **Ada-RefSR**, leveraging **AICG** to adaptively balance intrinsic SR fidelity with reference-guided enhancement, mitigating hallucinations.*
> > * ***Generalization.*** *AICG enables Ada-RefSR to generalize RefSR beyond narrow domains to diverse scenarios, maintaining robustness under varying reference alignment.*
> > * ***Efficiency.*** *Built on single-step diffusion, Ada-RefSR achieves over $30\times$ speedup compared to ReFIR and reduces fallback re-synthesis via dynamic correction of mismatched references. Additionally, the lightweight AICG mechanism constructs implicit correlations, which reduces the extra computational cost compared to explicit correlation modeling.*
>
> #### Revised Contributions (New Focused Structure):
>
> > * ***AICG: Adaptive Implicit Correlation Gating.*** *We propose **AICG**, a lightweight implicit correlation gating module that directly addresses a key challenge in RefSR: how to reliably use reference information to restore LQ inputs degraded by real-world artifacts. By reusing existing projections in the attention module and introducing only a few learnable summary tokens, AICG implicitly models LQ–Ref correlations while adding negligible computational overhead.*
> > * ***Ada-RefSR: Strong generalization, robustness, and speed.*** *Built upon AICG, **Ada-RefSR** achieves stable reference-based enhancement across diverse tasks and degradation scenarios. Its single-step diffusion design provides over $30\times$ speedup compared to multi-step RefSR baselines, enabling fast and robust SR in both aligned and mismatched reference conditions.*
>
> ---
>
> ### Conclusion Refinement
>
> To ensure the conclusion clearly reflects the methodological novelty emphasized in the rebuttal, we updated the final paragraph of the paper. The revision (1) places stronger emphasis on a **key RefSR challenge**—the unreliable utilization of reference information under real-world degradations, and (2) more explicitly highlights the **centrality and lightweight nature of AICG** in resolving this issue. Compared to the original version, the revised conclusion now better articulates how AICG enables reliable, efficient, and robust reference guidance, aligning the closing remarks with the main contributions of the method.

---

> ### Author Response · Authors · 2025-11-21
> **Response to weaknesses 2**
>
> We thank you for your insightful comment. We clarify that the efficiency of Ada-RefSR comes from **two complementary sources**: (1) the **implicit-correlation AICG gating** that avoids costly explicit token matching, and (2) the **one-step diffusion RefSR pipeline** used as the backbone. Both aspects are now explicitly discussed in the revised manuscript.
>
> ---
>
> ### 1. Efficiency of AICG vs. Explicit Gating (New Analysis)
>
> A major efficiency advantage of Ada-RefSR stems from its **implicit correlation–based gating** design.
>
> Relative to the vanilla Reference Attention (RA), **AICG introduces only 0.13% additional FLOPs** (see Appendix E for detailed derivations). In contrast, ReFIR-style *explicit correlation gating* requires computing and storing a dense token-to-token similarity map of size $L_{\text{src}} \times L_{\text{ref}}$, leading to roughly **16% overhead** over vanilla RA.
>
> This significant gap results from the fact that AICG relies on a small set of $M=16$ learnable summary tokens to model cross-image correlations implicitly. Explicit correlation methods instead incur quadratic complexity and memory cost due to the dense similarity computation. Appendix E includes the theoretical FLOPs breakdown and a direct numerical comparison.
>
> ---
>
> ### 2. Overall Runtime Efficiency (Clarified)
>
> As reported in **Table 3** of the revised manuscript, Ada-RefSR is approximately **30× faster** than SeeSR+ReFIR at $1024 \times 1024$ resolution, while introducing only about $\sim 1.9\times$ overhead relative to the single-image S3Diff baseline. This superior efficiency gain is attributed to (i) the lightweight reference pathway and (ii) the inherently efficient **one-step diffusion backbone**.
>
> ---
>
> ### 3. Model Size (New Clarification)
>
> We agree that parameter count merits clearer discussion. The reported numbers correspond to the **core Ada-RefSR model**, where the current implementation employs a full-capacity ReferenceNet symmetric to S3Diff, resulting in a total of 2678.89M parameters (1326.77M from the S3Diff backbone, 61.98M from the Reference Attention module, 0.20M from learnable summary tokens, and 1289.95M from ReferenceNet). For comparison, SeeSR+ReFIR uses 2039.83M parameters, while S3Diff alone has 1326.77M, making our core model roughly 2× larger than the S3Diff baseline.
>
> Importantly, despite having a larger parameter count, Ada-RefSR remains **substantially more efficient in runtime** (e.g., **30× faster** than SeeSR+ReFIR), owing to the **one-step diffusion architecture** and the negligible overhead of AICG.
>
> We also clarify that the larger model size is not inherent to our design: lightweight variants—such as compact or shallower ReferenceNet, token-pruned reference attention, or distilled reference encoders—can be readily incorporated and will be explored in future work.
>
> ---
>
> ### Revisions Added to the Manuscript
>
> To fully address the reviewer’s concern, we have added:
>
> * A new theoretical complexity analysis of AICG in **Appendix E**, including direct comparison with ReFIR-style explicit correlation gating.
> * New **Tables 5 and 6** summarizing FLOPs and runtime comparisons.
> * Updated discussion in **Section 4.4** clarifying model size and its relation to runtime efficiency (lines 480–482).
> * Additional explanation of AICG’s lightweight gating mechanism (lines 504–509).

---

> ### Author Response · Authors · 2025-11-21
> **Response to weaknesses 3**
>
> We thank you again for the helpful suggestion. In response, we have added additional visual comparisons focusing specifically on **complex, man-made textures**. These examples are drawn from the **CUFED5 dataset** and include WiFi logos, door frames, wall tiles, printed text, and fine fabric patterns. The newly added results are provided in **Appendix G** (see Figures 15 and 16 in the revised manuscript).
>
> Across these challenging cases, Ada-RefSR consistently preserves structural fidelity and accurately transfers fine-scale reference textures:
>
> * **WiFi symbol (CUFED5\_26):** SUPIR+ReFIR produces over-bright colors, and SeeSR+ReFIR introduces shape distortions, turning smooth circular boundaries into unnaturally sharp edges. Single-step SR baselines (PISA-SR, OSEDiff, S3Diff) fail to properly remove degradations. In contrast, **Ada-RefSR maintains the correct rounded geometry of the symbol and restores its soft boundary and color, closely matching both the HQ and the reference.**
> * **Artificial structures (door frames, tiles, logos, fabric):** Competing methods often yield hallucinated textures or lose geometric alignment. **Ada-RefSR more faithfully reconstructs repeating patterns, straight edges, and subtle material textures**, demonstrating stronger coherence with the reference and better adherence to the true content.
>
> We believe these results highlight two key advantages of our design:
> 1.  Our framework retains the **strong single-image SR prior** from the baseline model, enabling effective degradation removal; and
> 2.  The introduced reference-guided enhancement—implemented under our **trust but verify paradigm**—allows the model to incorporate reference details selectively and reliably.
> Together, these properties contribute to **robust generalization** across both natural scenes and artificial, structured textures.
>
> ---
>
> ### Revisions Added to the Manuscript
>
> To fully address the reviewer’s concern, we have added:
>
> * **Figures 15 and 16** with results on complex artificial textures.
> * A new **Appendix section G** (lines 1147–1169) discussing the behavior on man-made structures.

---

### Official Review · Reviewer_c7Cn · 2025-10-31

**Soundness:** 3
**Presentation:** 3
**Contribution:** 3
**Rating:** 6
**Confidence:** 5

**Summary:**

This paper presents Ada-RefSR, a single-step diffusion framework for reference-based super-resolution that follows a “Trust but Verify” strategy. The method first injects reference features through a reference-attention module and then uses an Adaptive Implicit Correlation Gating (AICG) mechanism to adaptively filter mismatched reference cues. Experimental results on multiple datasets show improved fidelity, perceptual quality, and efficiency compared with recent diffusion and RefSR methods.

**Strengths:**

- The paper clearly defines the problem of over- or under-reliance on reference images in diffusion-based SR and provides a coherent conceptual framework to address it.

- The proposed AICG module is lightweight and easily pluggable into existing backbones, enabling adaptive control of reference guidance without additional supervision.

- This paper provides clear visual evidence, such as attention maps, gating masks, and token visualizations, that help interpret the mechanism’s behavior.

**Weaknesses:**

- The main limitation lies in the novelty boundary of AICG. Its design using learnable tokens for implicit correlation modeling is conceptually close to DETR-style query or prototype aggregation, and the paper does not clearly explain how it fundamentally differs from those approaches.

- The paper also lacks comparisons with strong face-specific RefSR methods, which would help validate the generalization of Ada-RefSR in specialized domains.

- There is no detailed analysis of different reference scenarios, such as varying reference quality, domain gaps, or alignment errors, which would strengthen the claim of robustness under diverse reference conditions.

**Questions:**

See weaknesses.

---

> ### Author Response · Authors · 2025-11-21
> **Overall Response to Reviewer c7Cn**
>
> ### Overall Response
>
> We sincerely thank you for the positive assessment and for recognizing the effectiveness of our **"Trust but Verify"** framework. We have addressed the concerns regarding novelty, specialized comparisons, and robustness analysis with substantial revisions.
>
> #### Summary of Revisions:
>
> 1.  **Clarified AICG vs. DETR Distinction:** We have explicitly revised the manuscript to clarify that AICG tokens serve as **internal reliability estimators** for reference gating, fundamentally differing in purpose and mechanism from DETR's **object decoding slots**. (W1 response)
>
> 2.  **Added Specialized Comparisons:** We incorporated two strong face-specific baselines, **FaceMe** and **InstantRestore**. Ada-RefSR demonstrates superior fidelity and perceptual performance against these specialized methods (updated Table 1 and Fig. 18). (W2 response)
>
> 3.  **Validated Robustness to Alignment/Quality:** We introduced a new systematic stress-test section (**Appendix F**) with controlled Gaussian blur and affine misalignment experiments. This analysis confirms Ada-RefSR's stability and adaptive reference suppression under diverse degradation scenarios. (W3 response)

---

> ### Author Response · Authors · 2025-11-21
> **Response to weaknesses 1**
>
> Thank you for the insightful comment. We acknowledge that our original writing may have unintentionally implied a close connection between AICG and DETR. We would like to clarify that although AICG borrows the *high-level idea* of using learnable tokens, its purpose, mechanism, and position in the architecture are fundamentally different from DETR.
>
> First, **DETR queries serve as prediction slots** in a decoder and are directly responsible for object-level decoding. In contrast, **AICG’s learnable tokens are not prediction units**: they function purely as internal summarization carriers that aggregate dominant reference patterns to estimate *implicit reliability scores* for reference features.
>
> Second, **the computational role is different**. DETR performs cross-attention for decoding object instances. AICG, however, operates entirely inside the **Reference Attention module**: its learnable tokens first summarize reference keys, after which LQ queries attend to these compact summaries to compute token-wise gating signals. This two-stage implicit correlation process has no counterpart in DETR.
>
> Third, **the objectives differ fundamentally**. DETR targets object detection, whereas AICG is specifically designed for diffusion-based RefSR, where the primary challenge is to regulate reference feature injection under misalignment, noise, or severe viewpoint changes—conditions that DETR does not tackle.
>
> We appreciate the reviewer’s suggestion and have revised the manuscript to clearly distinguish AICG’s role from DETR and to better highlight its unique purpose in stabilizing reference usage for real-world RefSR.
>
> ---
>
> **Revision in the manuscript.**
> To avoid misunderstanding, we have updated the paragraph on Page 5, Lines 238-242 (revised manuscript). The original text:
>
> > *"Following the idea of using learnable queries in DETR (Carion et al., 2020), we employ learnable tokens to distill dominant reference modes into compact summaries that correlate with LQ queries to yield discriminative gating signals..."*
>
> has been revised to the following clearer and more accurate description:
>
> > *"While inspired by the high-level idea of learnable tokens in DETR (Carion et al., 2020), AICG differs fundamentally: instead of acting as decoding queries, our tokens summarize reference features to provide implicit, token-wise reliability estimates for gating. This design suppresses noise and mismatches while retaining essential structures."*

---

> ### Author Response · Authors · 2025-11-21
> **Response to weaknesses 2**
>
> ### Table 2: Comparison of different methods on image restoration metrics.
>
> | Method          |  PSNR (↑)  | SSIM (↑)  | LPIPS (↓) |  FID (↓)   | NIQE (↓)  | MUSIQ (↑)  | CLIP-IQA (↑) |
> | :-------------- | :--------: | :-------: | :-------: | :--------: | :-------: | :--------: | :----------: |
> | DMDNet          | 26.8993🥈  | 0.7472🥈  |  0.2316   |  56.6262   | 4.6030🥈  | 72.9495🥈  |    0.6395    |
> | CodeFormer      |  26.2677   |  0.7082   |  0.3798   |  99.6599   |  5.4105   |  63.9886   |    0.5192    |
> | **Faceme**          |  **26.6591**   |  **0.7245**   |  **0.2500**   |  **54.3278**   |  **4.7295**   |  **72.4144**   |  **0.6483 🥇**   |
> | **InstanceRestore** |  **26.2245**   |  **0.7350**   | **0.2070🥈**  | **51.2283🥈**  |  **5.9473**   |  **69.7209**   |    **0.5726**    |
> | **Ours**        | 27.1271 🥇 | 0.7523 🥇 | 0.1749 🥇 | 42.7042 🥇 | 4.4880 🥇 | 74.3828 🥇 |   0.6453🥈   |
> > **Table Note:** Arrows indicate the optimal trend (↑: higher is better; ↓: lower is better). Best results are marked with 🥇, and second best results are marked with 🥈.
>
>
>
> We thank you for highlighting the importance of evaluating Ada-RefSR in face-specific reference restoration settings. To provide a fair and comprehensive assessment, we additionally compare our method with two recent diffusion-based specialized approaches—**FaceMe** (Liu et al., 2025) and **InstantRestore** (Zhang et al., 2025)—as well as representative restoration baselines including DMDNet and CodeFormer. As shown in **Table 2**, Ada-RefSR achieves consistently competitive or superior results across both fidelity- and perception-oriented metrics.
>
> We further provide two possible explanations for these results:
>
> 1.  **Strong LQ priors from the single-step SR diffusion backbone.**
>     FaceMe and InstantRestore are trained from scratch and rely primarily on face-specific modeling, whereas **Ada-RefSR** is built upon a strong single-step SR diffusion backbone. This enables our model to better preserve reliable LQ structural priors while still benefiting from reference guidance.
>
> 2.  **Robustness to reference mismatch.**
>     FaceMe depends on explicit face localization, and InstantRestore directly injects reference features without a mismatch-mitigation mechanism. By contrast, our **AICG** module implicitly models LQ–Ref correlations and adaptively suppresses inconsistent cues, effectively handling imperfect, noisy, or partially misaligned face references. This leads to more stable reference utilization in challenging real-world cases.
>
> Overall, these additional experiments verify that Ada-RefSR generalizes well to specialized face-reference restoration, while maintaining fairness in comparison settings.
>
>
>
> ---
> **Revisions in the manuscript.**
>
> * We have added the new face-reference comparison results to Table 1 (Page 7, Lines 360–361 in the revised manuscript).
> * We updated Figure 18 in the supplementary material to include visual comparisons with FaceMe and InstantRestore (Page 26).
> * We expanded the experimental setup for face-reference evaluation and inserted citations to FaceMe and InstantRestore (Page 6, Line 310).
>
> ---
>
> The references for the specialized baselines mentioned are provided below:
>
> * **FaceMe**:
>     Siyu Liu, Zheng-Peng Duan, Jia OuYang, Jiayi Fu, Hyunhee Park, Zikun Liu, Chunle Guo, and
> Chongyi Li. *Faceme: Robust blind face restoration with personal identification.* In *Proceedings
> of the AAAI Conference on Artificial Intelligence*, 2025.
>
> * **InstantRestore**:
>     Howard Zhang, Yuval Alaluf, Sizhuo Ma, Achuta Kadambi, Jian Wang, and Kfir Aberman. *Instantrestore: Single-step personalized face restoration with shared-image attention.* In *Proceedings of the Special Interest Group on Computer Graphics and Interactive Techniques Conference Conference Papers, pp. 1–10*, 2025.

---

> ### Author Response · Authors · 2025-11-21
> **Response to weaknesses 3**
>
> Thank you again for this valuable suggestion. We fully agree that **robustness to diverse reference conditions** is a critical attribute for real-world RefSR. To address this, we conducted a systematic evaluation of Ada-RefSR under degraded, mismatched, and misaligned reference scenarios.
>
> ***
>
> ### 1. Experimental Setup: Controlled Stress-Testing
>
> Since the face benchmark references are typically well-aligned, we introduced two specific degradation protocols to rigorously test robustness (see **Fig. 11**):
>
> * **Gaussian Blur (Reference Quality):** We applied kernels of sizes $\{3, 5, 7, 9, 11\}$ to simulate defocused or low-resolution references.
> * **Affine Transformations (Alignment Error):** We applied five graded levels of rotation, translation, scaling, and corner distortion to simulate viewpoint inconsistencies.
>
> ***
>
> ### 2. Main Findings: Stability and Superiority
>
> Our quantitative and qualitative analyses reveal three key trends demonstrating the robustness of Ada-RefSR:
>
> * **Robustness to Misalignment (Fig. 12 (b)):** Under increasing affine distortion, Ada-RefSR demonstrates **remarkably stable FID and LPIPS curves**, whereas other reference-based baselines deteriorate rapidly.
> * **Robustness to Low Quality (Fig. 12 (a)):** Even under severe Gaussian blur, Ada-RefSR remains competitive with or superior to dedicated face RefSR models (e.g., InstantRestore, DMDNet, FaceMe).
> * **Artifact-Free Fallback (Fig. 14):** Qualitative results show that our model preserves identity and structure across all degradation levels. Crucially, when the reference becomes too degraded to be useful, our model naturally falls back to the single-image baseline performance (S3Diff) **without introducing artifacts**.
>
> ***
>
> ### 3. Mechanism Analysis: Source of Robustness
>
> To validate the source of this observed robustness, we analyzed the behavior of the **AICG** module:
>
> **Mechanism Visualization (Fig. 13).**
> We visualized the average AICG gating activations across network layers. As reference quality declines (via blur or misalignment), the gating activations **consistently decrease**. This confirms that AICG acts as an **adaptive filter**, automatically down-weighting or suppressing unreliable reference cues to prevent **negative transfer**.
>
> ***
>
> ### 4. Summary & Revisions
>
> These experiments confirm that Ada-RefSR follows a "**trust but verify**" paradigm: leveraging the reference when reliable and actively suppressing it when degraded. We have incorporated these analyses into the revised manuscript:
>
> * **New Section (Appendix F):** A detailed analysis of robustness under misalignment and low-quality references.
> * **Fig. 11 \& 14:** Visual examples of the degradation protocols and qualitative comparisons.
> * **Fig. 12 \& 13:** Quantitative performance curves and AICG activation visualizations.

---

### Official Review · Reviewer_cRgB · 2025-10-31

**Soundness:** 2
**Presentation:** 3
**Contribution:** 3
**Rating:** 6
**Confidence:** 5

**Summary:**

The paper proposes Ada-RefSR, a single-step reference-based diffusion framework for super-resolution. The method addresses the challenge of establishing reliable correspondences between low-quality (LQ) inputs and reference images, which is particularly difficult under severe degradations. To mitigate issues of over- or under-reliance on references seen in existing weighting-based fusion strategies, the authors adopt a “Trust but Verify” paradigm: the model leverages references when they align and suppresses them otherwise. The core component, Adaptive Implicit Correlation Gating (AICG), uses learnable summary tokens to capture dominant reference patterns and implicit correlations with LQ features. Integrated into the attention backbone, AICG adaptively regulates reference guidance and prevents erroneous feature fusion. Experiments on multiple datasets show that Ada-RefSR achieves a good balance of fidelity, naturalness, and efficiency, while maintaining robustness under varying reference alignment.

**Strengths:**

1. The proposed “Trust but Verify” perspective is an interesting and promising approach.

2. The authors conduct extensive experiments to validate the effectiveness of their method.

**Weaknesses:**

1. The key innovation of this paper is the Adaptive Implicit Correlation Gating (AICG) mechanism. Currently, the introduction lacks a critical figure illustrating the authors’ approach and the core differences compared with prior methods, which is highly important.

2. The authors need to explain why their method does not perform well on the WRSR dataset.

**Questions:**

No

---

> ### Author Response · Authors · 2025-11-21
> **Overall Response to Reviewer cRgB**
>
> Thank you for your positive assessment, particularly for highlighting the **“Trust but Verify”** approach and the comprehensive experiments. We have made the following concise revisions to address the core concerns regarding conceptual clarity and dataset analysis:
>
> ### Summary of Revisions and Clarifications:
>
> 1.  **Conceptual Clarity of AICG (New Figure):**
>     * We added a new **Figure 2** to conceptually contrast AICG with prior gating strategies (PFStorer, ReFIR).
>     * We clarified that AICG's novelty lies in its **implicit correlation estimation** using summary tokens, which provides more stable, adaptive, and efficient gating than explicit token matching, especially under real-world noise.
>
> 2.  **Analysis of WRSR Performance:**
>     * We clarified that the metric pattern on the WRSR dataset reflects a deliberate **fidelity–perception balance**, not a failure.
>     * Our method successfully removes real-world degradations and achieves substantial gains over the diffusion baseline (S3Diff) on all perceptual metrics, demonstrating effective reference utilization while avoiding the artifact-prone nature of many traditional and multi-step RefSR methods.

---

> ### Author Response · Authors · 2025-11-21
> **Response to weaknesses 1**
>
> Thank you for the question. The key distinction of our method lies in the fundamentally different way we design the gating mechanism. To make this clear, we added **Fig. 2** in the revised manuscript, which directly contrasts PFStorer, ReFIR, and our AICG. Briefly:
>
> 1.  **PFStorer** uses a single global learnable vector to weight warped reference features. Since this vector is shared across all images, it cannot reflect the varying alignment quality between the LQ and reference images, limiting its applicability in real-world scenarios.
>
> 2.  **ReFIR** computes an explicit token-wise similarity map between LQ and reference features. However, such raw correlations are highly sensitive to noise and long-tailed token distributions. Mild but abundant tokens often dominate the similarity matrix, suppressing the few highly informative ones. This results in unstable gating and can lead to artifact-prone reference injection.
>
> 3.  **AICG (ours)** employs a small set of learnable *aggregation tokens* that first summarize the dominant reference patterns before interacting with LQ queries. This implicit correlation estimation avoids noise-dominated explicit similarity maps, requires no manually tuned coefficients (e.g., the mixing factor $\lambda$ in ReFIR), and performs correlation in a compact low-dimensional space.
>
>
> **We further note that although AICG may appear more complex in structure (as visualized in Fig. 2), its computational overhead is in fact significantly lower.**
> As detailed in Appendix E of the revised manuscript, AICG introduces only **0.13%** additional computation compared to vanilla reference attention, whereas ReFIR incurs a substantially larger **16%** overhead due to explicit token–token correlation and the storage of large similarity matrices. This comparison highlights that AICG is not only more robust but also far more computationally efficient than explicit correlation–based gating.
>
> Overall, the newly added figure and clarification better illustrate why our implicit, real-world–oriented gating mechanism provides more stable, adaptive, and efficient reference utilization than prior approaches.
>
> ---
>
> **Revision in the manuscript.**
> In the revised manuscript, we have explicitly addressed this concern by:
> 1.  Adding Fig. 2 to conceptually compare the gating strategies.
> 2.  Adding the citation for Fig. 2 in Lines 74-76 of the main text.

---

> ### Author Response · Authors · 2025-11-21
> **Response to weaknesses 2**
>
> ### Table 1: The performance evaluation on the WRSR dataset.
>
> | Method            |  PSNR (↑)   |  SSIM (↑)  | LPIPS (↓)  |   FID (↓)    |  NIQE (↓)   |  MUSIQ (↑)  | CLIPIQA (↑) |
> | :---------------- | :---------: | :--------: | :--------: | :----------: | :---------: | :---------: | :---------: |
> | **LQ**            | **23.1260** | **0.5944** | **0.7891** | **154.1796** | **11.4410** | **18.9054** | **0.2641**  |
> | C2-Matching\*     | 22.8766 🥈  |   0.5784   |   0.7471   |   153.1606   |   9.0121    |   19.9772   |   0.1527    |
> | DATSR\*           | 22.9017 🥇  | 0.5820 🥈  |   0.7583   |   153.5259   |   9.3999    |   20.1124   |   0.1613    |
> | Real-ESRGAN-S1    |   22.1491   | 0.5974 🥇  |   0.3631   |   97.9142    |  3.6976 🥈  |   64.5176   |   0.6076    |
> | S3Diff-S1         |   21.9122   |   0.5620   |   0.3542   |   63.8237    |   4.4171    |   63.8237   |   0.6149    |
> | OSEDiff-S1        |   21.3263   |   0.5603   |   0.3261   |   71.5148    |   3.7364    | 71.1935 🥈  |  0.6899 🥈  |
> | PISA-SR-S1        |   21.8813   |   0.5682   | 0.3383 🥈  |   62.9958    |   4.0359    |   69.5187   |   0.6407    |
> | SUPIR+ReFIR\*-S50 |   21.0134   |   0.5378   |   0.3949   |   76.4993    |  3.6407 🥇  |   70.1242   |   0.6219    |
> | SeeSR+ReFIR\*-S50 |   21.8334   |   0.5673   |   0.3435   |  61.9597 🥈  |   3.8919    | 72.1520 🥇  |  0.7120 🥇  |
> | Ours-S1\*         |   21.9722   |   0.5777   | 0.3061 🥇  |  53.2811 🥇  |   3.7429    |   69.8608   |   0.6612    |
>
> ---
>
> **Note**: Methods with \* use an external reference image. Metrics are averaged over the WRSR test set. The best result is marked with 🥇, and the second best result is marked with 🥈.
>
> ## Main reply
>
> Thank you for pointing out this issue. We would like to clarify that the results on **WRSR** do not indicate a failure of our method. Instead, they reflect a fundamental **fidelity–perception balance** inherent in real-world RefSR. WRSR stresses both real-world degradation removal and reference-faithful detail recovery, and different classes of baselines sit at opposite extremes of this spectrum. Our method occupies a balanced middle ground, which explains the metric distribution observed in Table 1.
>
> 1.  **Traditional RefSR (C2-Matching, DATSR): High PSNR/SSIM but no degradation removal.**
>     These methods assume synthetic bicubic degradation and thus fail to handle real-world degradations in WRSR. As shown in **Fig. 15 and Fig. 16**, they largely preserve the corrupted LQ structures. This behavior leads to **inflated PSNR/SSIM**, consistent with the fact that the raw LQ input already achieves the highest PSNR (Table 1). Their scores therefore do not reflect successful restoration, but rather insufficient modification of the degraded input.
>
> 2.  **Our method vs. diffusion baseline (S3Diff): Clear, consistent improvement.**
>     Our approach is built upon S3Diff. Incorporating reference attention + AICG brings **strong gains across all No-Reference (NR) and perceptual metrics** and visible restoration improvements. This demonstrates that our method effectively leverages reference cues while removing real-world degradations, unlike regression-based RefSR models.
>
> 3.  **Large-scale multi-step RefSR (SeeSR+ReFIR, SUPIR+ReFIR): High NR scores but hallucination-prone.**
>     These models tend to optimize NR metrics (e.g., NIQE) by relying on generative priors rather than the reference image, often leading to hallucinated details unrelated to the source.
>     In contrast, our **AICG** module constrains reference usage and prevents such hallucination, yielding better fidelity to the GT and the reference image.
>
> 4.  **Summary: Why our performance pattern on WRSR is expected and reasonable.**
>     WRSR requires simultaneously (a) removing unknown real-world degradations and (b) leveraging reference images without hallucination.
>     * *Traditional RefSR methods* maximize PSNR by keeping degradations intact;
>     * *large-scale diffusion methods* maximize NR scores but still face the hallucination problem;
>     * **our method strikes a principled balance**—recovering faithful structure while maintaining perceptual realism.
>     Thus, the metric pattern on WRSR reflects this balance rather than a failure of the method.
>
> We hope this explanation clarifies the behavior of our method on WRSR and the underlying trade-offs that shape the evaluation results.
>
> **Revision in the manuscript.** We have updated the manuscript (Lines 371-372) to explicitly note that on the WRSR dataset, our method achieves a balanced trade-off between degradation removal and faithful reference utilization, reflecting its fidelity–perception balance under real-world conditions.

---

### Author Response · Authors · 2025-11-21
**Summary of Revisions and Responses**

We thank all reviewers for their constructive and insightful feedback.
In response, we have significantly expanded and refined both the main paper and the appendix. All newly added or modified content is highlighted in **green** in the revised manuscript. Below we summarize the key changes:

1.  **New Conceptual Comparison Figure (Reviewer cRgB, weakness 1).**
    We introduce a new **Figure 2** to clearly contrast our AICG (implicit correlation based gating) with prior representative strategies, demonstrating AICG's efficiency and robustness.

2. **Clarification of WRSR Performance (Reviewer cRgB, weakness 2).**
    We updated the manuscript (Lines 371-372) to note that our performance on WRSR reflects a **fidelity–perception balance** under real-world conditions, clarifying the trade-off between degradation removal and faithful reference utilization.

3. **Clarification of AICG vs. DETR Distinction (Reviewer c7Cn, weakness 1).**
    We revised the manuscript (Page 5, Lines 238-242) to clarify that AICG tokens are **internal summarization carriers** for reliability gating, fundamentally differing from DETR's **object decoding slots**.

4.  **Additional Comparisons with Specialized Methods (Reviewer c7Cn, weakness 2).**
    We incorporated two recent face-specific approaches, **FaceMe** and **InstantRestore**, into quantitative (**Updated Table 1**) and qualitative (**Updated Figure 18**) comparisons, confirming Ada-RefSR's competitiveness.

5.  **Robustness Under Varying Degradation (Reviewer c7Cn, weakness 3).**
    We added comprehensive controlled experiments in **Appendix F** (New Figures 11–14) evaluating Ada-RefSR under Gaussian blur and affine misalignment. These results demonstrate its adaptive handling of imperfect references and AICG's robust behavior.

6.  **Motivation and Contributions Clarified (Reviewer Fy7H, weakness 1).**
    We revised the **Abstract**, **Introduction**, and **Conclusion** to clearly articulate the novelty of **AICG** in solving the unresolved challenge of reliable reference usage under real-world degradations.

7.  **Additional Computational Complexity Analysis (Reviewer Fy7H, weakness 2).**
    We expanded the complexity discussion in **Appendix E** and **Section 4.4** with a detailed FLOPs analysis, detailing model size and AICG’s minimal overhead.

8.  **New Visual Results on Complex Textures (Reviewer Fy7H, weakness 3).**
    We added new visual comparisons on **complex, man-made textures** in **Appendix G (Figures 15 and 16)**, confirming Ada-RefSR’s strong generalization to structured scenes.

9.  **Appendix Overview Added and Figure Reorganization.**
    We added an Appendix Overview and moved original Figures 11–13 to **Appendix H** (Figures 17–19) to improve narrative flow and readability.

These revisions substantially strengthen our claims regarding robustness, efficiency, and generalization, and we believe they comprehensively address the reviewers' concerns.

---

### Author Response · Authors · 2025-12-01
**Summary of Rebuttal Improvements for the AC**

**We provide this brief summary to assist the AC in evaluating our rebuttal, and we sincerely thank the AC for their time and consideration.** As the reviewers have not yet responded, we respectfully highlight the main clarifications and improvements made during the rebuttal period.

---
### **Our Main Contributions for RefSR task**

1. **Addresses a key challenge in Real-world RefSR.**
    We address an important but underexplored challenge in RefSR: real-world degradations make direct LQ–Reference matching unreliable. Our work focuses on **how to reliably use reference information to restore LQ inputs degraded by real-world artifacts**.

2. **AICG, lightweight and robust reference regulation.**
    We introduce **Adaptive Implicit Correlation Gating (AICG)**, a highly lightweight module that adds **negligible computational overhead**. Without extra supervision, AICG adaptively suppresses mismatched reference signals and strengthens reliable ones, leading to more robust guidance than explicit correlation-based gating.

3. **Extending RefSR to broader scenarios with strong efficiency.**
    Built upon AICG, **Ada-RefSR** naturally generalizes to multiple reference settings—including face restoration, category-level reference, and scene-level SR. Thanks to its one-step diffusion design and the efficiency of AICG, Ada-RefSR delivers high-quality results while being **~30× faster** than existing multi-step reference-based SR pipelines.


---

### **Responses to Reviewers' Core Concerns**

- **Implicit correlation vs. prior explicit methods (reviewer cRgB).**
    We added a new Figure 2 to illustrate why our implicit correlation gating is more robust across varied reference conditions and significantly more efficient than explicit correlation-based gating used in prior RefSR methods.

- **Fundamental distinction from DETR-style mechanisms (reviewer c7Cn).**
    We revised the manuscript to directly clarify the essential difference between AICG and DETR-style decoding mechanisms. While AICG employs learnable tokens, its **formulation, computational role, and objective** are fundamentally different: AICG summarizes reference features to estimate reliability and gate reference usage under real-world degradations—capabilities that fall outside the scope of DETR.

- **Clarifying the contribution for RefSR task (reviewer Fy7H).**
    We refined the abstract and introduction to clearly highlight that AICG is specifically designed to **adaptively regulate reference usage under real-world degradations**, providing stable and noise-robust reference guidance—an aspect insufficiently addressed by prior RefSR methods.

---

### **Additional Experiments Requested by Reviewers**

To further substantiate the robustness of our approach, we have performed extensive new experiments and visualizations that decisively address reviewers' feedback:

1. Comparisons with strong face-specific baselines (FaceMe, InstantRestore), demonstrating strong performance in the face domain. (requested by reviewer c7Cn)
2. Controlled robustness studies (blur, misalignment), further confirming AICG’s adaptive and reliable reference utilization under diverse reference conditions.  (requested by reviewer c7Cn)
3. Visualizations on complex, man-made textures, showing effective mitigation of hallucination artifacts and demonstrating that our method can reliably transfer detailed textures from the reference.  (requested by reviewer Fy7H)

These additions provide stronger evidence for adaptive and reliable reference utilization across diverse reference conditions.

---
We confirm that all reviewer concerns and suggestions, including the detailed complexity analysis of AICG and the specific performance explanation on the WRSR dataset, have been fully and meticulously incorporated into the revised manuscript.

We sincerely invite the AC to consider these substantial improvements, as we believe the paper now offers a robust and efficient solution to the RefSR community.

---

### Meta-Review · Area_Chair_s1pM · 2025-12-28

**Summary:**

Three reviews are received on this paper.

Reviewer cRgB has concerns on the unclear presentation of the key innovations and the explanation on the experiments.

Reviewer c7Cn has concerns on the limited novelty, the lack of comparisons with strong face-specific RefSR methods, and the lack of detailed analysis of different reference scenarios.

Reviewer Fy7H’s concerns focus on the limited novelty, insufficient discussions and experiments on efficiency and model size, and the generalization performance of the model.

**Reviewer Concerns:**

None of the reviewers provide feedback on the authors’ rebuttal. The authors’ rebuttal is very detailed. The AC thinks that Reviewer cRgB’s concerns are addressed. For the other two reviewers, some concerns on the novelty and comparison may remain. However, the AC agrees that this paper solves a somewhat new task of real-world face super-resolution with reference images. Therefore, it can be accepted.

**Reviewer Scores:**

The ACs think that Reviewer cRgB and Reviewer c7Cn will keep their score of 6. For Reviewer Fy7H, considering that the authors have provided very detailed response, the ACs think that he/she may also raise the score to 6.

---

### Decision · Program_Chairs · 2026-01-26

Accept (Poster)